# Machine Learning Recognizes Stages of Parkinson’s Disease Using Magnetic Resonance Imaging

**DOI:** 10.3390/s24248152

**Published:** 2024-12-20

**Authors:** Artur Chudzik

**Affiliations:** Faculty of Computer Science, Polish-Japanese Academy of Information Technology, 86 Koszykowa Street, 02-008 Warsaw, Poland; artur.chudzik@pjwstk.edu.pl

**Keywords:** machine learning (ML), Parkinson’s disease (PD), magnetic resonance imaging (MRI), Euclidean distance, Cosine distance

## Abstract

Neurodegenerative diseases (NDs), such as Alzheimer’s disease (AD) and Parkinson’s disease (PD), are debilitating conditions that affect millions worldwide, and the number of cases is expected to rise significantly in the coming years. Because early detection is crucial for effective intervention strategies, this study investigates whether the structural analysis of selected brain regions, including volumes and their spatial relationships obtained from regular T1-weighted MRI scans (*N* = 168, PPMI database), can model stages of PD using standard machine learning (ML) techniques. Thus, diverse ML models, including Logistic Regression, Random Forest, Support Vector Classifier, and Rough Sets, were trained and evaluated. Models used volumes, Euclidean, and Cosine distances of subcortical brain structures relative to the thalamus to differentiate among control (HC), prodromal (PR), and PD groups. Based on three separate experiments, the Logistic Regression approach was optimal, providing low feature complexity and strong predictive performance (accuracy: 85%, precision: 88%, recall: 85%) in PD-stage recognition. Using interpretable metrics, such as the volume- and centroid-based spatial distances, models achieved high diagnostic accuracy, presenting a promising framework for early-stage PD identification based on MRI scans.

## 1. Introduction

There is a common agreement that the human brain exhibits small-world topology, characterized by high local clustering and specific path lengths between regions [1,2]. This structure is characterized by strongly interconnected neurons forming modules, and these modules are linked together by a smaller number of long-distance connections [3]. Thus, as in every topological structure, the relationship between physical distance and connectivity in networks exhibits fundamental trade-offs among cost, signal propagation, and processing capacity [4].

Therefore, geometric distance also introduces fundamental communication delays between nodes in the brain. Specifically, in brain networks, the speed at which signals propagate between regions is limited by the physical properties of the connections, such as the axon length and state of myelination [5]. These delays are particularly relevant in the context of synchronization and various neural oscillations used to coordinate communication across brain regions. Long-distance communication often occurs at slower frequencies (<25 Hz, such as theta or delta oscillations), which are better suited for transmitting information over longer distances without losing synchronization. Short-range communication, on the other hand, typically occurs at higher frequencies (>40 Hz, such as gamma oscillations), where speed is more critical, but distance is shorter [6].

As the brain ages, long-distance connections tend to deteriorate faster than short-range connections [7,8]. This “disconnected brain” phenomenon, characterized by age-related reductions in structural and functional connectivity, has been shown to account for significant declines in executive functions and processing speed. Specifically, structural connectivity changes, particularly in long-distance white matter tracts linking the prefrontal cortex to other regions, explain up to 82.5% of the decline in the executive function over time [8]. Furthermore, this phenomenon is particularly evident in the structural connectome, where hubs—regions with long-distance, high-capacity connections, and high metabolic rates—are more vulnerable to aging effects than peripheral connections [7]. These age-related changes in hub connections reflect biological vulnerability and also contribute to cognitive decline, such as reduced processing speed.

Notably, this deterioration is significantly more pronounced in neurodegenerative diseases (NDs), such as Alzheimer’s disease (AD), Parkinson’s disease (PD), and frontotemporal dementia (FTD), impairing the brain’s ability to integrate information across different regions [9]. These disturbances in neuronal structures are caused by the degradation of both neuronal tissue and white matter, leading to the abnormal organization of selected brain regions.

Therefore, there is a group of studies that explore the geometric and topological properties of brain networks in neurodegenerative diseases. For example, Supekar et al. (2008) observed disrupted small-world properties and local connectivity, particularly in the hippocampus, in AD [10]. Furthermore, Dubbelink et al. (2014) found that PD patients exhibit decreased local clustering and network decentralization, which progress over time and correlate with declining motor and cognitive function [11]. Moreover, Garg et al. (2016) introduced a novel Geometry Networks framework to analyze the cortical surface geometry in PD, showing significant differences in homology features between PD and control groups [12].

Additionally, Chakraborty et al. (2020) presented that textural, morphological, and statistical features of selected brain structures vary among healthy controls, prodromal PD, and Parkinson’s patients [13]. Following this conclusion, Suo et al. (2021) found that the topological organization of gray matter networks is disrupted in early-stage Parkinson’s disease and that there is potential for applications to early imaging diagnosis using MRI [14].

These studies collectively demonstrate the potential of network analysis and geometric approaches in understanding neurodegenerative diseases and developing biomarkers for diagnosis and progression monitoring.

However, multiple gaps remain that need to be addressed for the practical implementation of these findings. First, there is no widely accepted optimal approach for defining nodes and edges that can be used for calculations, which leads to inconsistencies across studies in how brain regions and their connections are identified and modeled. Additionally, studies on the connectome often incorporate additional metrics, such as neuropsychological tests as co-variates, to enhance the accuracy, sensitivity, and specificity of the findings.

Clinical questionnaires are often an integral part of the clinical diagnosis process, and comparing their performance against MRI-based metrics is therefore essential to establish their usefulness. In general, the tests used in clinical practice have been subjected to rigorous validation. A common choice includes MoCA, which has proven to be more effective than some other cognitive tests, such as the Mini-Mental State Examination (MMSE), especially in the context of Parkinson’s disease [15,16]. However, this introduces variability that can make the selection of the best features for ML-aided prognosis difficult. This is because MoCA’s performance can vary across diverse cultural and educational backgrounds, potentially leading to the misinterpretation of results [17]. Moreover, individuals with lower education levels may score lower on the MoCA, not necessarily due to cognitive impairment but due to the test’s design, which might favor those with higher educational and cultural backgrounds [18]. Another noteworthy concern is its susceptibility to practice effects, especially between the first and second administrations. This could potentially skew results and require consideration in clinical interpretation [19].

Furthermore, parameters determining the shape and structure of deep brain regions, which are critical for understanding neurodegenerative processes, are often influenced by the resolution limitations of MRI scanners, leading to potential inaccuracies in network characterization. These challenges created the need for the development of alternative approaches for defining nodes and edges in network models that can be applied consistently across different research settings.

### 1.1. Disrupted Networks in Parkinson’s Disease

Because the brain is composed of many regions, we will now consider the main pathways affected in Parkinson’s disease. PD is characterized by the degeneration of dopamine-producing neurons in the substantia nigra (SN), a key midbrain structure within the basal ganglia, which is essential for movement control. The substantia nigra produces dopamine (DA), a neurotransmitter that is crucial for smooth and coordinated muscle activity [20] (Figure 1). Dopamine deficiency disrupts the basal ganglia’s role in regulating movement, leading to the motor symptoms of PD [21].

In PD, dopamine loss particularly affects D1 and D2 dopamine receptors, which play distinct roles in movement pathways. The D1 receptors are involved in the “GO” (direct) pathway, facilitating movement initiation, while D2 receptors participate in the “noGO” (indirect) pathway, inhibiting unnecessary movements. Reduced dopamine levels suppress the “GO” pathway and overactivate the “noGO” pathway [20].

This imbalance increases inhibitory output from the globus pallidus externus (GPe) to the subthalamic nucleus (STN), decreasing STN excitation of the globus pallidus internus (GPi). At the same time, reduced “GO” pathway activity leads to less inhibition of the GPi, causing it to exert stronger inhibition on the thalamus. This heightened inhibition weakens thalamic signals to the motor cortex (M1 and SMA), resulting in slowed and impaired movements [20].

As PD progresses, degeneration begins in the substantia nigra and spreads to other structures involved in movement control. This progression manifests as symptoms, like tremors, rigidity, bradykinesia, and postural instability. While the exact causes of neuronal loss remain unclear, they likely involve a combination of genetic and environmental factors, complicating the identification of a single biomarker for PD.

One of the possible hypotheses suggests that neuronal loss is the effect of impaired function of the glymphatic system (GS) [22,23,24]. The GS is a “waste clearance” mechanism in the brain that plays a key role in regulating the exchange of cerebrospinal fluid (CSF) and interstitial fluid (ISF). It is responsible for the removal of toxic metabolites, thus maintaining brain homeostasis.

Dysfunction of this system has been implicated in various neurodegenerative diseases, including PD [25]. Recent findings suggest that glymphatic dysfunction in PD could be linked to impaired CSF dynamics, further influencing the basal ganglia’s role in motor control [26]. The reduced clearance of metabolic waste may contribute to neuroinflammation and degeneration in the substantia nigra and other interconnected regions, triggering dopamine loss.

### 1.2. Purpose of the Study

This study evaluates whether data from standard MRI scans can be used to calculate relationships and whether volumes of selected brain structures can serve as straightforward parameters for a structural relationship analysis.

Therefore, this study focuses on interpretable metrics, namely the volumes and the spatial distances between the centroids of selected brain structures. These metrics should provide a simple representation of the brain’s topology, from which basic mathematical operations can be conducted to model parameters of connectivity and relationships.

First, a volumetric analysis will be performed on all structures. Following this, the centroids of selected structures will be used to calculate the Euclidean distance to the thalamus, providing a measure of their spatial separation. To capture more complex spatial relationships, a second metric, the Cosine distance, will be introduced, which accounts for the orientation of the structures relative to the thalamus. As a result, each structure will be characterized by three parameters: volume, Euclidean distance from the thalamus, and Cosine distance from the thalamus. The goal is to find the best features for the ML-aided diagnosis of PD.

## 2. Methods

Data used in the preparation of this article were obtained from the Parkinson’s Progression Markers Initiative (PPMI) database (www.ppmi-info.org (accessed 15 July 2024)). The PPMI is an observational clinical study that aims to identify biomarkers of Parkinson’s disease progression by assessing multiple cohorts to develop the largest collection of clinical, imaging, and biologic specimens ever created in the Parkinson’s community. PPMI data and specimens are made accessible through websites to academic and industry researchers to perform verification studies of PD biomarkers.

### 2.1. MRI Dataset

MRI scans were acquired from control (*n* = 56), prodromal (*n* = 56), and Parkinson’s disease (*n* = 56) groups during baseline (BL) visits, resulting in a total of 168 participants. All scans were performed with a 3.0 Tesla MRI scanner using sagittal acquisition planes. The imaging protocol employed either a 3D T1-weighted sequence or a 3D MPRAGE GRAPPA sequence, with a slice thickness of 1.0 mm (Table 1). These scans were used for volumetric analysis and brain segmentation to investigate volumetric differences across the groups.

The PPMI is a multi-site study, and there was an uneven distribution of participants from each group across scanners. This can result in increased classification accuracy due to scanner differences and must be considered carefully. The heterogeneity in MRI machine characteristics, such as variations in software, sensors, algorithms, and scan protocols, provide inter- and intra- site variations that can lead to the failure of downstream and machine learning (ML) processes, creating a challenge in the analysis [27].

Therefore, FreeSurfer was used in this study, as it was demonstrated to show good test–retest reliability across scanner manufacturers and across field strengths [28,29]. Moreover, despite not being used in this study, data harmonization can further improve the accuracy and generalizability of new AI models, allowing for more accurate results and better generalization to new datasets [27].

### 2.2. Data Preprocessing

Cortical reconstruction and volumetric segmentation were performed with the FreeSurfer image analysis suite, which is documented and freely available for download online (http://surfer.nmr.mgh.harvard.edu/ accessed 15 July 2024). The technical details of these procedures are described in prior publications [28,29,30,31,32,33,34,35,36,37,38,39,40,41,42,43,44,45,46,47,48].

### 2.3. Segmentation of Brain Regions

The process of segmentation includes motion correction and the averaging [35] of multiple volumetric T1-weighted images (when more than one is available). However, in this study, no participants had repeated scans, and this method was not applied. The segmentation process also includes the removal of non-brain tissue using a hybrid watershed/surface deformation procedure [30], automated Talairach transformation, segmentation of the subcortical white matter and deep gray matter volumetric structures (including the hippocampus, amygdala, caudate, putamen, ventricles) [32,45], intensity normalization [37], tessellation of the gray matter white matter boundary, automated topology correction [36,46], and surface deformation following intensity gradients to optimally place the gray/white and gray/cerebrospinal fluid borders at the location where the greatest shift in intensity defines the transition to the other tissue class [33,38,41]. Once the cortical models are complete, a number of deformable procedures can be performed for further data processing and analysis including surface inflation [39], registration to a spherical atlas, which is based on individual cortical folding patterns to match cortical geometry across subjects [44], parcellation of the cerebral cortex into units with respect to the gyral and sulcal structure [48], and the creation of a variety of surface-based data, including maps of the curvature and sulcal depth. This method uses both intensity and continuity information from the entire three-dimensional MR volume in segmentation and deformation procedures to produce representations of cortical thickness, calculated as the closest distance from the gray/white boundary to the gray/CSF boundary at each vertex on the tessellated surface [47]. The maps are created using spatial intensity gradients across tissue classes and are therefore not simply reliant on the absolute signal intensity. The maps produced are not restricted to the voxel resolution of the original data and thus are capable of detecting submillimeter differences between groups. Procedures for the measurement of cortical thickness have been validated against histological analyses [42] and manual measurements [40,41]. FreeSurfer morphometric procedures have been demonstrated to show good test–retest reliability across scanner manufacturers and across field strengths [28,29].

While FreeSurfer requires relatively long operating times, the program runs and segments data automatically with a single line of code (Script 1) [49].

**Script 1.** FreeSurfer command used for MRI segmentation.

recon-all -i {input_mri_path} -s {study_name} -all -sd {output_path}

### 2.4. Quality Control

This study implemented a quality control pipeline to address the concerns regarding the reconstruction accuracy. This workflow systematically evaluated the outputs of recon-all using automated and visual inspection tools. First, the pipeline scanned all study directories to identify subjects and their respective recon-all logs. It checked for critical errors and filtered relevant output to avoid false positives. For visual QC, the pipeline generated key anatomical screenshots (sagittal, coronal, axial) overlaid with segmentation (aseg.mgz) using Freeview, allowing the general inspection of surface and segmentation accuracy.

Additionally, aseg.stats files were parsed to summarize volumetric data, ensuring that structures are segmented correctly and data were complete. Reconstruction relied strictly on automatic segmentation, without the manual adjustment of recognized brain areas. Subjects with missing or empty files were flagged for review and excluded from the study (nine scans of Parkinson’s disease patients). The review document included the status of each subject, error logs, availability of statistical data, and scanner model metadata.

### 2.5. Feature Extraction

The selection of structures for this analysis was guided by their relevance to both the pathophysiology of Parkinson’s disease and the practical limitations of available segmentation tools. The final set included the thalamus, hippocampus, amygdala, caudate, putamen, pallidum, accumbens area, ventral diencephalon, cerebellum, fourth ventricle, and cerebrospinal fluid (CSF). These structures were chosen because of their direct or indirect involvement in motor control, cognitive function, and neurodegenerative processes.

The thalamus serves as a relay center for basal ganglia output to the cortex [50], while the caudate, putamen, and pallidum form core components of the basal ganglia, playing critical roles in movement regulation [51]. The hippocampus and amygdala, part of the limbic system, contribute to the cognitive and emotional aspects of PD [52]. The accumbens area, as part of the ventral striatum, integrates reward and motivational signals, which can also be affected in PD [53]. The ventral diencephalon includes the hypothalamus and was found to be an important region for MRI-based classification models in the context of MCI and dementia [54]. The cerebellum fine-tunes motor activity and coordinates movements, often compensating for basal ganglia dysfunction [55]. Additionally, the fourth ventricle and CSF were included to coarsely assess parts of the glymphatic system [56], of which impairment has been linked to neuroinflammation in neurodegenerative diseases.

Understandably, it would be optimal to focus on all the brain regions that were identified in the connectivity graph, including the substantia nigra parameters. However, this study uses only the basic FreeSurfer atlas, which has a limited number of recognized structures to ensure easier reproducibility while maintaining a focus on key anatomical regions involved in PD pathology. Future research should focus on additional atlases that may provide additional and relevant parameters for the described method and additional relevant structures.

All metrics described in this research were calculated using the voxel data segmented by FreeSurfer for each deep brain structure and cerebrospinal fluid (CSF), utilizing the following FreeSurfer labels (Figure 2): Thalamus (label 10 plus label 49), Left-Hippocampus (17), Right-Hippocampus (53), Left-Amygdala (18), Right-Amygdala (54), Left-Caudate (11), Right-Caudate (50), Left-Putamen (12), Right-Putamen (51), Left-Pallidum (13), Right-Pallidum (52), Left-Accumbens-area (26), Right-Accumbens-area (58), Left-VentralDC (28), Right-VentralDC (60), 4th-Ventricle (15), Left-Cerebellum (7 plus 8), Right-Cerebellum (46 plus 47), and CSF (24).

### 2.6. Volumetric Analysis

For all selected structures, FreeSurfer provided explicit volume measurements, and these volumes were used directly in the analysis. All volumes of brain structures and CSF were normalized per participant using their thalamus volume. The idea of normalizing brain volumes to a reference is a technique to control individual variability in brain size or shape. This allows for comparisons between different groups (such as HC, PD, and PR) to focus on relative differences in the volumes of the structures of interest, rather than being confounded by overall brain size differences between individuals.

Data normalization is an important approach to reduce inter-subject variability in group studies, but care must be taken when choosing a normalization strategy to avoid introducing confounds or artefacts into the data. In this study, normalization based on the thalamus was chosen to focus on subcortical structures directly related to Parkinson’s disease pathology. While the intracranial volume (ICV) is widely used, it includes cortical structures that are not relevant to this study.

However, it was found that among subcortical regions, the thalamus consistently presents the highest positive correlation with the ICV, irrespective of the diagnosis of neurodegenerative disease (AD) [57]. Moreover, the thalamus normalization method was recently investigated by Zhang et al. (2024) who observed that it improves the detectability of hypoperfusion in Alzheimer’s disease and lessens the artefacts caused by the commonly used strategy of normalization using the global mean [58]. Therefore, this study follows proposed strategies and applies normalization using thalamus volumetry.

### 2.7. Spatial Metric Computations

In FreeSurfer, after automatic subcortical segmentation, each voxel was labeled according to the structure it belonged to (e.g., left amygdala, CSF), and its coordinates were transformed into the RAS (Right-Anterior-Superior) space to obtain *x*, *y*, and *z* values.

Consequently, each structure could be represented as a cloud of points in three-dimensional space. The centroid (central point) of that structure was calculated simply by averaging the x, y, and z coordinates of all voxels within a given structure.

Once the centroids were established, they were used to calculate both the Euclidean distance (in millimeters) and Cosine distance (unitless) relative to the centroid of the thalamus, which served as the reference point. This method allowed for spatial measurements and comparisons between the structures, which were normalized to account for individual differences (Figure 3). The selection of Euclidean and Cosine distances is an approach that uses both the established effectiveness of Euclidean metrics in structural neuroimaging and the potential of Cosine metrics for multidimensional analysis [59,60].

The Euclidean distance is a classic metric for measuring spatial relationships in 3D space. Persistent homology features obtained from Euclidean distance matrices presented effectiveness in detecting geometric changes [59]. Hence, this metric aligns well with the need to measure spatial displacements and changes in anatomical regions and is presented to be useful in the context of ND detection [59,61].

The Euclidean distance can be considered a specific case of the Minkowski distance when the parameter *p* = 2. The general formula for the Minkowski distance is given by the following:$$D_{minkowski}\left(x,y\right)={\left(\sum_{i=1}^{n}{\left|x_{i}-y_{i}\right|}^{p}\right)}^{\frac{1}{p}}$$

When *p* = 2, this simplifies to the formula for the Euclidean distance:$$D_{euclidean}\left(x,y\right)=\sqrt{\sum_{i=1}^{n}{\left(x_{i}-y_{i}\right)}^{2}}$$
where

$x=\left(x_{1},x_{2},\ldots ,\;x_{n}\right)$ and $y=\left(y,y_{2},\ldots ,\;y_{n}\right)$ are two points in n-dimensional space.$x_{i},\;y_{i}$ are the coordinates of the points *x* and *y* along the *i*-th dimension.

An additional metric, the Cosine distance, evaluates angular relationships between vectors in high-dimensional spaces, making it suitable for analyzing patterns of spatial arrangements. It captures how “aligned” different regions or features are relative to a reference. A PubMed search highlights the lack of studies that applied Cosine distance in the context of brain structures in Parkinson’s research. However, this metric is successfully applied in various biomedical data analyses [60,62,63] and thus will be investigated experimentally in this study.

The Cosine distance ranges from 0 (when the vectors are identical) to 2 (when the vectors are diametrically opposed in direction). It is commonly used in text mining and other applications involving high-dimensional sparse data, as it measures orientation rather than magnitude.

The Cosine distance is defined as follows:$$D_{cosine}\left(x,y\right)=1-\frac{\sum_{i=1}^{n}x_{i}y_{i}}{\sqrt{\sum_{i=1}^{n}x_{i}^{2}}\cdot \sqrt{\sum_{i=1}^{n}y_{i}^{2}}\;}$$
where

$\sum_{i=1}^{n}x_{i}y_{i}$ is the dot product of x and y.$\sqrt{\sum_{i=1}^{n}x_{i}^{2}}$ is the Euclidean norm (magnitude) of x.$\sqrt{\sum_{i=1}^{n}y_{i}^{2}}$ is the Euclidean norm (magnitude) of y.

These norms are multiplied in the denominator to normalize the dot product. Multiplication ensures that the similarity is based on the angle between the vectors and irrespective of their magnitudes.

### 2.8. Statistical Analysis

Descriptive statistics, including the mean and standard deviation, were calculated for each group across all relevant parameters. Group differences were evaluated using ANOVA as a global test to identify overall differences among groups. Post-hoc pairwise comparisons were subsequently conducted using Welch’s *t*-tests (independent *t*-tests with unequal variance assumptions) for the following group pairs: HC vs. PR, HC vs. PD, and PR vs. PD.

Additionally, the “Sex” variable was coded as binary (male [M] = 0, female [F] = 1) and analyzed across groups using chi-squared tests to account for its categorical nature.

To control for multiple comparisons and reduce the risk of Type I error, p-values were adjusted using the false discovery rate (FDR) correction. This method was selected for its balance between statistical rigor and sensitivity in high-dimensional datasets.

Statistics were calculated for four categories of data (demographics and clinical scores, volumetric analysis, cosine distances, and Euclidean distances analysis). Results were compiled into tables. All analyses were conducted using Python 3.10.12, with Pandas 2.2.2, SciPy 1.14.1, and Seaborn 0.13 libraries [64,65,66].

### 2.9. Machine Learning

For data modeling, four machine learning models were used, namely Logistic Regression (LR), Random Forest (RF), Support Vector Classifier (SVC), and Rough Sets (RS). The choice of LR, RF, and SVC algorithms is supported by their earlier usefulness in the context of neurodegenerative disease detection [67]. In this study, LR, RF, and SVC were implemented in Python using the scikit-learn 1.5 (sklearn) library [68].

The Rough Set approach was included as an additional validation technique due to its demonstrated effectiveness in granular computing with rough-set rules [69]. This approach followed the Pawlak model for Rough Sets, which uses rule extraction and reasoning about data [70,71,72]. Parameters were optimized to minimize ambiguity and maximize coverage. Rough Sets were modeled in Rough Set Exploration System 2.2 (RSES) [70,72].

Existing research concluded that the optimum results are achieved if 20 to 30% of the data are used for model testing and the remaining 70 to 80% of the data are used to train the model [73]. Thus, a 30/70 train–test split was applied to ensure that 70% of the data were used for training and 30% for testing. This approach was chosen to maintain consistency across classification tasks and prioritize the comparability of results.

Additionally, hyperparameter tuning for LR, RF, and SVC was performed using a grid search with cross-validation to optimize model performance. Figure 4 shows the workflow with the processing pipeline.

Initially, the dataset included 57 parameters (thalamus volume, three parameters per each of the remaining 18 brain structures, along with sex and age). To prevent overfitting and reduce noise, the dataset was divided into three sub-datasets: (1) healthy controls and prodromal, (2) HC and Parkinson’s, and (3) prodromal and Parkinson’s.

The goal was to find the best set of training data to improve classification, but the information in excess can cause noise. This noise leads the model to learn random patterns that do not improve its ability to sort images correctly. This excess of information is called overfitting and is a well-known problem in machine learning. Therefore, the best-ranked features can be used as a strategy to avoid overfitting and improve and optimize the model [74].

This feature-selection method follows the findings of a study by Imran et al. (2018), where the authors applied five feature selection techniques namely the Gain Ratio, Kruskal–Wallis Test, Random Forest Variable Importance, RELIEF, and Symmetrical Uncertainty, along with four classification algorithms (K-Nearest Neighbor, Logistic Regression, Random forest, and Support Vector machine) on the PD dataset [75]. They found that Random Forest Variable Importance is one of the best impactful feature ranking techniques.

Thus, in the present study within each sub-dataset, Random Forest was employed first to vote for the most important features specific to each classification task.

## 3. Results

This study included 168 participants who were divided into three groups: 56 healthy controls (HCs), 56 prodromal (PR), and 56 Parkinson’s disease (PD) patients. All participants were selected based on the availability of high-quality 3D T1-weighted or MPRAGE GRAPPA scans from baseline visits in the Parkinson’s Progression Markers Initiative (PPMI) database (Table 1). The groups were matched for age and gender for balanced comparisons across groups.

### 3.1. MRI Scans and Sensors

All MRI scans were acquired at a field strength of 3.0 Tesla. The majority of scans used 3D T1-weighted sequences, such as MPRAGE GRAPPA for Siemens scanners and equivalent sequences for other manufacturers (Table 2).

The common acquisition parameters were as follows:Acquisition Plane: sagittalSlice Thickness: 1.0 mmVoxel Size: 1 × 1 × 1 mm^3^Field of View (FOV): approximately 256 × 256 mm^2^Matrix Size: approximately 256 × 256Repetition Time (TR): 4–2300 msEcho Time (TE): 2–3 msInversion Time (TI): 400–900 ms (for sequences with inversion recovery)Flip Angle: 9–15 degreesAcceleration Techniques: GRAPPA or similar parallel imaging techniques where applicable

Sequence MPRAGE GRAPPA was used predominantly with Siemens scanners (TrioTim and Verio models), providing high-resolution images with good gray–white matter contrast. 3D T1-weighted sequences were used with GE, Philips, and Toshiba scanners, optimized for anatomical detail.

Given the multi-site and multi-scanner nature of the PPMI dataset, several steps were taken to address potential variability due to different scanners and acquisition protocols, as described in the Quality Control.

Initially, 65 Parkinson’s disease patients met the inclusion criteria for this study. However, after data processing with FreeSurfer 7.4.1, only 56 scans were successfully segmented and analyzed. The remaining participants were excluded due to processing failures, which could be attributed to issues, such as motion artifacts, incomplete scans, or segmentation errors. To maintain balanced classes for machine learning modeling, the prodromal and control groups were also limited to 56 random participants each.

### 3.2. Energy Consumption

Each MRI scan required approximately 3 h to process using a machine equipped with a 13th generation Intel Core i9-13900KF processor and 128 GB of RAM. This configuration adequately matched the computational demands of FreeSurfer. Calculations were performed in six parallel threads to speed up the process of segmentation of many MRI scans. The total energy consumption for these calculations was as follows: $\frac{168\;scans\times 3\;h/scan}{6\;threads}*\sim 280\;W\;\approx 23.5\;kWh$.

### 3.3. Demographics and Clinical Scores

Table 3 presents the demographic and clinical characteristics of the participants across the control, Parkinson’s disease, and prodromal groups, along with statistical comparisons between the groups.

The “Sex” variable was coded as binary (male [M] = 0, female [F] = 1) and analyzed across groups using chi-squared tests to account for its categorical nature. The sex distribution between groups was relatively balanced, with the PD group having a higher percentage of females (43%) compared to the control group (32%) and the prodromal group (25%). Statistical analysis showed no significant differences in sex distribution between the groups (χ^2^ = 4.071, *p* = 0.260).

The mean age of participants in the HC group was 61.54 years (SD = 11.49), the PD group had a mean age of 61.84 years (SD = 10.99), and the PR group had a higher mean age of 66.34 years (SD = 6.16). However, there was no significant difference in age among the groups (ANOVA *p* = 0.065).

Motor symptom severity, as assessed based on the UPDRS3 scores, was significantly different across groups (ANOVA *p* < 0.001). The PD group presented the highest UPDRS3 scores, with a mean of 23.89 (SD = 11.47), compared to the control group, which had a mean UPDRS3 score of 0.79 (SD = 1.67), and the prodromal group, which had an intermediate mean score of 4.86 (SD = 5.91). Post-hoc pairwise comparisons using Welch’s *t*-tests revealed that the differences were statistically significant between all groups (all *p* < 0.001).

### 3.4. Volumetric Analysis

The volumetric analysis of subcortical brain structures revealed several significant differences between the groups, as shown in Table 4.

**Thalamus:** No significant differences were found in the thalamus volume among the groups (Figure 5, ANOVA *p* = 0.108).

**Hippocampus:** No significant differences were observed in the volumes of the left or right hippocampus among the groups (left hippocampus ANOVA *p* = 0.065; right hippocampus ANOVA *p* = 0.589).

**Amygdala:** The left amygdala showed significant differences among the groups (ANOVA *p* = 0.001). Post-hoc comparisons revealed that both the PD and PR groups had significantly smaller left amygdala volumes compared to the HC group (HC vs. PD *p* = 0.002, *t* = 3.928; HC vs. PR *p* = 0.002, *t* = 4.010), while no significant difference was found between the PD and PR groups (*p* = 0.799, *t* = −0.466). This indicates that atrophy in the left amygdala may occur early in neurodegeneration. No significant differences were observed in the right amygdala (ANOVA *p* = 0.945).

**Caudate Nucleus:** The left caudate showed significant differences among the groups (ANOVA *p* = 0.034). A post-hoc analysis indicated that the PD group had a significantly smaller left caudate volume compared to the HC group (HC vs. PD *p* = 0.015, *t* = 3.138). However, no significant differences were found in the HC vs. PR (*p* = 0.096, *t* = 2.197) or PD vs. PR (*p* = 0.465, *t* = −1.053) comparisons. This suggests that atrophy in the left caudate may be associated with PD but not necessarily present in the prodromal stage. No significant differences were observed in the right caudate (ANOVA *p* = 0.200).

**Putamen:** Both the left and right putamen volumes showed significant differences among the groups (left ANOVA *p* = 0.002; right ANOVA *p* = 0.016). Post-hoc comparisons revealed that both the PD and PR groups had significantly smaller putamen volumes compared to the HC group (left putamen HC vs. PD *p* = 0.002, *t* = 3.869; HC vs. PR *p* = 0.001, *t* = 4.114; right putamen HC vs. PD *p* = 0.035, *t* = 0.665; HC vs. PR *p* = 0.006, *t* = 3.528). No significant differences were found between the PD and PR groups (left *p* = 0.984; right *p* = 0.707). These results suggest that putamen atrophy is detectable in the early stages of PD and may be present even in the prodromal phase.

**Globus Pallidus:** No significant differences were observed in the volumes of the left or right globus pallidus among the groups (left ANOVA *p* = 0.167; right ANOVA *p* = 0.945). Therefore, changes in the globus pallidus volume may not be a distinguishing factor in early PD or its prodromal stage.

**Accumbens Area:** The left accumbens area showed significant differences among the groups (ANOVA *p* = 0.001). Both the PD and PR groups had significantly smaller left accumbens volumes compared to the HC group (HC vs. PD *p* = 0.001, *t* = 4.374; HC vs. PR *p* = 0.001, *t* = 4.354), while no significant difference was observed between the PD and PR groups (*p* = 0.859). This suggests that atrophy in the left accumbens area may be an early indicator of PD. No significant differences were observed in the right accumbens area (ANOVA *p* = 0.347).

**Ventral Diencephalon:** The left ventral diencephalon showed significant differences among the groups (ANOVA *p* = 0.003). The PD group had a significantly smaller left ventral diencephalon volume compared to the HC group (HC vs. PD *p* = 0.002, *t* = 3.959). The difference between HC and PR approached significance (*p* = 0.051, *t* = 2.566) but did not reach the threshold. No significant difference was found between the PD and PR groups (*p* = 0.227, *t* = −1.640). This suggests that atrophy in the left ventral diencephalon is associated with PD. No significant differences were observed in the right ventral diencephalon (ANOVA *p* = 0.133).

**Cerebellum:** No significant differences were observed among the groups for the volumes of the left and right cerebellum (left ANOVA *p* = 0.258; right ANOVA *p* = 0.220).

**CSF:** No significant differences were observed among the groups for the volumes of cerebrospinal fluid (ANOVA *p* = 0.177).

### 3.5. Euclidean Distance Analysis

The volumetric analysis of subcortical brain Euclidean distances revealed several significant differences between the groups, as shown in Table 5.

**Thalamus:** The thalamus is the reference point in this analysis; therefore, Euclidean distances are measured from the thalamus to other structures. No direct Euclidean distance involving the thalamus itself is provided for comparison.

**Hippocampus:** The Euclidean distance between the thalamus and the left hippocampus showed significant differences among the groups (Figure 6, ANOVA *p* = 0.015). Post-hoc comparisons revealed that both the PD and PR groups had significantly shorter distances compared to the HC group (HC vs. PD *p* = 0.015, *t* = 3.140; HC vs. PR *p* = 0.019, *t* = 2.981), while no significant difference was found between the PD and PR groups (*p* = 0.674). This suggests that spatial changes in the left hippocampus may occur early in neurodegeneration. No significant differences were observed in the Euclidean distances between the thalamus and the right hippocampus among the groups (ANOVA *p* = 0.984).

**Amygdala:** No significant differences were found in the Euclidean distances between the thalamus and the left amygdala among the groups (ANOVA *p* = 0.184). No significant differences were observed in the Euclidean distances between the thalamus and the right amygdala among the groups (ANOVA *p* = 0.250).

**Caudate Nucleus:** No significant differences were found in the Euclidean distances between the thalamus and the left caudate among the groups (ANOVA *p* = 0.966). The Euclidean distance between the thalamus and the right caudate showed significant differences among the groups (ANOVA *p* = 0.015). Post-hoc comparisons revealed that both the PD and PR groups had significantly shorter distances compared to the HC group (HC vs. PD *p* = 0.011, *t* = −3.256; HC vs. PR *p* = 0.017, *t* = −3.042), while no significant difference was found between the PD and PR groups (*p* = 0.941). This suggests that spatial reorganization of the right caudate nucleus may occur early in PD.

**Putamen:** No significant differences were observed in the Euclidean distances between the thalamus and the left putamen among the groups (ANOVA *p* = 0.945). Significant differences were found in the Euclidean distances between the thalamus and the right putamen among the groups (ANOVA *p* = 0.017). The PD group had a significantly longer distance compared to the HC group (HC vs. PD *p* = 0.006, *t* = −3.524). The difference between HC and PR groups was not significant (*p* = 0.108) nor was the difference between PD and PR groups (*p* = 0.343). This indicates that spatial alterations in the right putamen may be associated with PD.

**Globus Pallidus:** No significant differences were observed in the Euclidean distances between the thalamus and the left pallidum among the groups (ANOVA *p* = 0.721). Significant differences were found in the Euclidean distances between the thalamus and the right pallidum among the groups (ANOVA *p* = 0.000). Both the PD and PR groups had significantly longer distances compared to the HC group (HC vs. PD *p* = 0.000, *t* = −4.801; HC vs. PR *p* = 0.004, *t* = −3.641), while no significant difference was found between the PD and PR groups (*p* = 0.542). This suggests that spatial changes in the right globus pallidus may be an early indicator of PD.

**Accumbens Area:** No significant differences were observed in the Euclidean distances between the thalamus and the left accumbens among the groups (ANOVA *p* = 0.945). No significant differences were found in the Euclidean distances between the thalamus and the right accumbens among the groups (ANOVA *p* = 0.710).

**Ventral Diencephalon:** The Euclidean distance between the thalamus and the left ventral diencephalon showed significant differences among the groups (ANOVA *p* = 0.008). The PR group had a significantly shorter distance compared to the HC group (HC vs. PR *p* = 0.001, *t* = 4.058), indicating that spatial changes may begin in the prodromal stage. The difference between the HC and PD groups approached significance (*p* = 0.072, *t* = 2.363), but no significant difference was found between the PD and PR groups (*p* = 0.389). No significant differences were observed in the Euclidean distances between the thalamus and the right ventral diencephalon among the groups (ANOVA *p* = 0.171).

**Cerebellum:** Significant differences were found in the Euclidean distances between the thalamus and the left cerebellum among the groups (ANOVA *p* = 0.029). The PD group had a significantly shorter distance compared to the HC group (HC vs. PD *p* = 0.011, *t* = 3.252), suggesting cerebellar reorganization in PD. No significant differences were observed between the HC and PR groups (*p* = 0.244) or between the PD and PR groups (*p* = 0.206). No significant differences were observed in the Euclidean distances between the thalamus and the right cerebellum among the groups (ANOVA *p* = 0.107).

**CSF:** The Euclidean distance between the thalamus and CSF approached significance among the groups (ANOVA *p* = 0.051). Post-hoc comparisons revealed that the PR group had a significantly greater distance compared to the HC group (HC vs. PR *p* = 0.018, *t* = −3.019), indicating potential changes in CSF distribution in the prodromal stage. No significant differences were observed between the HC and PD groups (*p* = 0.095) or between the PD and PR groups (*p* = 0.674).

### 3.6. Cosine Distance Analysis

The volumetric analysis of subcortical brain Cosine distances revealed several significant differences between the groups, as shown in Table 6.

**Thalamus:** The thalamus is the reference point in this analysis; therefore, Cosine distances are measured from the thalamus to other structures. No direct Cosine distance involving the thalamus itself is provided for comparison.

**Hippocampus:** The Cosine distance between the thalamus and the left hippocampus showed significant differences among the groups (Figure 7, ANOVA *p* = 0.009). Post-hoc comparisons revealed that both the PD and PR groups had significantly smaller Cosine distances compared to the HC group (HC vs. PD *p* = 0.017, *t* = 3.067; HC vs. PR *p* = 0.011, *t* = 3.273), while no significant difference was found between the PD and PR groups (*p* = 0.984). This suggests that changes in the relative orientation of the left hippocampus may occur early in the prodromal stage and persist through the stages of the disease (Figure 2). No significant differences were observed in the Cosine distances between the thalamus and the right hippocampus among the groups (ANOVA *p* = 0.721).

**Amygdala:** No significant differences were observed in the Cosine distances between the thalamus and the left amygdala among the groups (ANOVA *p* = 0.171). No significant differences were observed in the Cosine distances between the thalamus and the right amygdala among the groups (ANOVA *p* = 0.910).

**Caudate Nucleus:** The Cosine distance between the thalamus and the left caudate showed significant differences among the groups (ANOVA *p* = 0.011). Post-hoc comparisons revealed that the PR group had a significantly smaller Cosine distance compared to the HC group (HC vs. PR *p* = 0.001, *t* = 4.240). The difference between PD and PR groups approached significance (*p* = 0.051, *t* = 2.574), while no significant difference was found between the HC and PD groups (*p* = 0.540). This suggests that reorientation of the left caudate relative to the thalamus may occur during the prodromal stage. No significant differences were observed in the Cosine distances between the thalamus and the right caudate among the groups (ANOVA *p* = 0.112).

**Putamen:** The Cosine distance between the thalamus and the left putamen showed significant differences among the groups (ANOVA *p* = 0.004). Post-hoc comparisons revealed that the PR group had a significantly smaller Cosine distance compared to the HC group (HC vs. PR *p* = 0.000, *t* = 4.801). No significant differences were found between the HC and PD groups (*p* = 0.257) or between the PD and PR groups (*p* = 0.073). These findings suggest that changes in the orientation of the left putamen may begin in the prodromal stage. No significant differences were observed in the Cosine distances between the thalamus and the right putamen among the groups (ANOVA *p* = 0.322).

**Globus Pallidus:** The Cosine distance between the thalamus and the left globus pallidus showed significant differences among the groups (ANOVA *p* = 0.017). Post-hoc comparisons revealed that the PR group had a significantly smaller Cosine distance compared to the HC group (HC vs. PR *p* = 0.001, *t* = 4.163). No significant differences were found between the HC and PD groups (*p* = 0.411) or between the PD and PR groups (*p* = 0.108). This suggests that orientation shifts in the left globus pallidus may occur during the prodromal stage. No significant differences were observed in the Cosine distances between the thalamus and the right globus pallidus among the groups (ANOVA *p* = 0.494).

**Accumbens Area:** No significant differences were observed in the Cosine distances between the thalamus and the left accumbens area among the groups (ANOVA *p* = 0.128). No significant differences were observed in the Cosine distances between the thalamus and the right accumbens area among the groups (ANOVA *p* = 0.250).

**Ventral Diencephalon:** No significant differences were observed in the Cosine distances between the thalamus and the left ventral diencephalon among the groups (ANOVA *p* = 0.220). No significant differences were observed in the Cosine distances between the thalamus and the right ventral diencephalon among the groups (ANOVA *p* = 0.945).

**Cerebellum:** No significant differences were observed in the Cosine distances between the thalamus and the left cerebellum among the groups (ANOVA *p* = 0.343). No significant differences were observed in the Cosine distances between the thalamus and the right cerebellum among the groups (ANOVA *p* = 0.242).

**CSF:** No significant differences were observed in the Cosine distances between the thalamus and the CSF among the groups (ANOVA *p* = 0.342).

### 3.7. Machine Learning

#### 3.7.1. Experiment 1: Classification of Healthy Controls vs. Parkinson’s Patients

The goal of the first experiment was to distinguish between healthy controls (HC) and Parkinson’s patients (PDs) using selected features of volumetric and spatial relationships of deep structures relative to the thalamus.

The most informative variables, identified through preselection using a Random Forest classifier, included 11 features (Euclidean distances of Right-Pallidum, Right-Putamen, Right-Caudate, Left-Hippocampus; Cosine distances of Left-VentralDC, Left-Putamen, Left-Hippocampus, Left-Pallidum; Volumes of Left-Amygdala, Left-Caudate, Left-Accumbens).

Three standard machine learning models were evaluated: Random Forest, Support Vector Classifier (SVC), and Logistic Regression. Results were further validated using the Rough Set approach.

The Random Forest model, optimized with a maximum depth of 3 and 75 estimators, achieved an accuracy of 71%, showing balanced precision and recall. However, it performed with slightly lower scores than the other models.

The SVC, with a tuning of C = 10 and gamma = 0.01, demonstrated higher performance with an accuracy of 79% and equally strong precision and recall scores.

Finally, the Logistic Regression model, optimized with C = 1.0, achieved better than SVC’s performance, with 85% accuracy, and the same precision and recall scores of 85%. In the case of Logistic Regression, three controls were misclassified as PD, and two PD patients were misclassified as controls, showing some limitations in precision (Figure 8).

Table 7 presents a detailed comparison of models used in this experiment. Overall, the Logistic Regression model outperformed SVC and Random Forest, achieving the highest ROC (area = 0.89). The confusion matrices showed that these models correctly classified most participants.

#### 3.7.2. Validation of Experiment 1 Using Rough-Set Rules

In the independent workflow using fuzzy rough sets to classify healthy controls and Parkinson’s disease patients, a total of 182 rules were generated. The model demonstrated strong performance with an overall accuracy of 82%. The true positive rate was 0.82 for healthy controls and 0.81 for Parkinson’s patients (coverage = 1.0). The overall performance was high, with 14 out of 17 HCs and 13 out of 16 PDs correctly classified.

For instance, one significant rule identified participants as Parkinson’s patients if the Cosine distance of the left ventral diencephalon (Left-VentralDC) was less than 0.1056, the Euclidean distance of the right caudate nucleus exceeded 29.8441, and the normalized volume of the left accumbens area was below 0.03125. This rule was used in accurately describing 13 Parkinson’s cases.

Conversely, another key rule classified individuals as healthy controls if the Euclidean distance of the left hippocampus exceeded 30.8871 and the Euclidean distance of the right-pallidum-normalized volume was below 25.504. This rule covered 12 healthy controls.

#### 3.7.3. Experiment 2: Classification of Healthy Controls vs. Prodromal Patients

The goal of the second experiment was to distinguish between healthy controls (HCs) and prodromal Parkinson’s patients (PRs) using the selected features of volumetric and spatial relationships of deep structures and CSF relative to the thalamus.

The most informative variables, identified through preselection using a Random Forest classifier, included 16 features (Cosine distances of Left-Putamen, Left-Caudate, Left-Pallidum, Right-Pallidum, Left-VentralDC, Left-Hippocampus, Right-Caudate, Right-Putamen; Euclidean distances of Left-VentralDC, Left-Hippocampus, Right-Cerebellum, Right-Caudate; Volumes of Left-Accumbens, CSF, Left-Putamen, Left-Amygdala).

The Logistic Regression model, with hyperparameters tuned to C = 0.1 and max_iter = 100, appeared as the top performer with an accuracy of 85%. This model presented a precision of 89% and both a recall and F1 score of 85%.

The confusion matrix showed that the Logistic Regression model perfectly identified all prodromal cases. However, it misclassified five controls as prodromal (Figure 9). This presents slightly lower specificity when distinguishing healthy controls from prodromal cases. Despite this, the model presented a strong ability to detect prodromal subjects, driven by features, such as the spatial relationships of the left putamen and globus pallidus relative to the thalamus. Table 8 presents a detailed comparison of models used in this experiment.

#### 3.7.4. Validation of Experiment 2 Using Rough-Set Rules

An independent experiment using fuzzy rough sets was conducted to classify healthy controls versus prodromal Parkinson’s patients, generating a total of 185 rules. The model achieved total accuracy of 88%, with a true positive rate of 0.87 for HCs and 0.89 for PR groups. These results present the model’s high performance in this task.

Several key rules were generated based on the combination of spatial and volumetric features. For example, one rule classified a participant as a healthy control if the normalized volume of the left amygdala exceeded 0.11165, the Cosine distance of the left putamen was greater than 0.1851, and the Euclidean distance of the right cerebellum exceeded 58.2086. This rule was able to accurately classify 18 healthy controls.

Conversely, another critical rule identified a participant as prodromal if the normalized volume of the left amygdala was below 0.11165, the Cosine distance of the left putamen was less than 0.1851, and the normalized volume of cerebrospinal fluid (CSF) exceeded 0.07115. This rule successfully classified 15 prodromal Parkinson’s cases.

There were still a few misclassifications, regarding the confusion matrix, where two out of 15 healthy controls were classified as prodromal and two out of 18 prodromal cases were classified as healthy controls. However, the model presented overall high accuracy. This indicates that while the model is effective, there remains some overlap in feature characteristics between the groups that can occasionally lead to misclassification.

#### 3.7.5. Experiment 3: Classification of Parkinson’s vs. Prodromal Patients

The goal of the third experiment was to distinguish between fully developed Parkinson’s patients (PDs) and prodromal Parkinson’s patients (PRs) using selected features of volumetric and spatial relationships of deep structures relative to the thalamus.

The most informative variables, identified through preselection using a Random Forest classifier, included 17 features (Cosine distances of Left-Putamen, Right-Accumbens, Left-Accumbens, Left-Cerebellum, Left-Amygdala, Left-Caudate, Right-Putamen, Right-Amygdala, 4th-Ventricle, Right-Caudate, Right-Cerebellum; Euclidean distance of Right-Cerebellum, Left-Hippocampus; Volumes of Left-Cerebellum, Left-Putamen, Left-VentralDC; and Age).

Feature importance analysis using the Random Forest model revealed that age had a low importance score (0.027439) and was only included in the classification model for the “PARKINSON” vs. “PRODROMAL” task. It was not utilized in other classification tasks, such as “CONTROL” vs. “PARKINSON” or “CONTROL” vs. “PRODROMAL”. This suggests that age did not play a significant role in driving the differences observed across groups.

The Logistic Regression model, with hyperparameters optimized to C = 0.1 and max_iter = 100, achieved an accuracy of 71% in distinguishing between prodromal and Parkinson’s subjects. The model demonstrated a precision of 72%, a recall of 71%, and an F1 score of 70%.

The confusion matrix showed that the model correctly identified 14 prodromal cases but misclassified 7 Parkinson’s patients as prodromal (Figure 10). This indicates a challenge in distinguishing between the two stages of the disease, particularly in classifying Parkinson’s patients. While the model had a high recall rate for identifying prodromal cases, it struggled with correctly identifying Parkinson’s cases.

Despite good but slightly lower overall accuracy, the model’s performance highlights the challenge of distinguishing between these two stages of the disease. The selected features, particularly the spatial relationships of the left caudate nucleus and cerebellum, played a critical role in classification, though further refinement of the model or additional features may be necessary to improve precision, particularly for identifying Parkinson’s patients. Table 9 presents a detailed comparison of models used in this experiment.

#### 3.7.6. Validation of Experiment 3 Using Rough-Set Rules

An additional validation was conducted using fuzzy rough sets, which generated 271 rules to classify prodromal and Parkinson’s disease cases. The model achieved a total accuracy of 66.7%, with a true positive rate of 0.63 for Parkinson’s patients and 0.71 for prodromal cases. Although the accuracy was lower compared to previous experiments, these results still offer valuable insights into the transition from prodromal to fully developed Parkinson’s disease.

Among those generated, one rule classified participants as prodromal if their age was greater than 63.5 years, the Cosine distance of the left caudate nucleus was below 0.0561, and the normalized volume of the left ventral diencephalon (Left-VentralDC) exceeded 0.26895. This rule accurately classified 12 prodromal cases, suggesting that certain volumetric and spatial characteristics, in combination with age, may be early indicators of Parkinson’s disease onset.

Conversely, another rule identified participants as having Parkinson’s disease if the Euclidean distance of the left hippocampus was below 30.1443, the Cosine distance of the left caudate nucleus exceeded 0.0561, the Euclidean distance of the right cerebellum was less than 58.8355, and the Cosine distance of the right cerebellum was below 0.9872. This rule classified 9 Parkinson’s patients.

The model faced challenges, particularly with distinguishing between patients with fully developed Parkinson’s disease and prodromal cases. However, the generated rules provided interpretable insights into the classification process.

#### 3.7.7. Ensemble Technique Needs External Validation

From a pragmatic perspective, it is feasible to merge the three best-performing models into a single workflow capable of receiving a single input and providing a unified diagnostic output. This approach employs an ensemble technique that aggregates the probabilistic outputs of individual models. By combining outputs from models trained on distinct classification tasks (HC vs. PD, HC vs. PR, PR vs. PD), aggregated probabilities for each diagnostic category—control, prodromal, and Parkinson’s disease—can be derived.

In practice, the ensemble calculates the final prediction by selecting the class with the highest summed probability. Additionally, confidence levels can be categorized into tiers, such as high, moderate, low, or extremely low, ensuring that the output reflects both the predicted diagnosis and the certainty of the model. This methodology enhances robustness and interpretability, making it better suited for real-world clinical applications.

However, this ensemble technique has not been validated on an independent dataset, which represents a limitation of this study. It is crucial to evaluate the ensemble on a completely separate test set that has not been used in training any of the models. Since the three models share overlapping training datasets (e.g., HC samples used in both HC vs. PD and HC vs. PR tasks), their probabilistic outputs may not be independent when validated on the same data. For future research, an independent test set with balanced samples of HC, PR, and PD categories should be used, ensuring no overlap with the training data. Such a dataset should also account for class imbalances common in real-world scenarios.

## 4. Discussion

This study demonstrates the utility of volumetric and spatial relationships of brain structures in distinguishing among healthy controls, prodromal, and Parkinson’s disease patients. Importantly, this study highlights the value of both Euclidean and Cosine distances, together with normalized volumetric data, to assess brain structure changes in PD.

The results demonstrate that changes in the volumes and spatial positioning of selected structures can serve as valuable markers for the early detection of PD. By normalizing brain structure volumes relative to the thalamus, models accounted for individual differences for clearer comparisons across healthy, prodromal, and PD groups. Specifically, the combination of volumetric data and spatial metrics proved to be a strong foundation for ML-based classification of the disease stages.

Interestingly, the Logistic Regression model consistently performed well across all experiments, with the highest accuracy in distinguishing both prodromal and PD cases from healthy controls (accuracy: HC vs. PD = 85%, HC vs. PR = 85%, PR vs. PD = 71%). This suggests that relationships between features, such as the volumes of key structures, like the hippocampus and amygdala, combined with their relative spatial orientations, are robust indicators of PD.

The fuzzy rough set approach achieved comparable results, indicating its potential as an alternative method for analyzing the spatial relationships and volumetric features of brain structures in Parkinson’s disease. For instance, in the HC vs. PR classification task, the fuzzy rough set model achieved an accuracy of 88%, surpassing the Logistic Regression model’s accuracy of 85%. Similarly, in the HC vs. PD classification, the fuzzy rough set model also performed well with an accuracy of 82%, close to the Logistic Regression model’s 85%.

On the other hand, models, such as Support Vector Classifier (SVC) and Random Forest, were also effective, but slightly less consistent across tasks, particularly in distinguishing between prodromal and Parkinson’s cases.

From a mechanistic perspective, disruptions in deep gray matter structures, such as the globus pallidus or the hippocampus, are linked to the dopaminergic degeneration characteristic of PD. The volumetric reductions and altered spatial relationships could reflect the progressive nature of these neurodegenerative changes. As a result, these findings align with other research showing the sensitivity of structural MRI and machine learning models in identifying subtle brain alterations associated with PD.

Overlapping features across experiments—such as Left-Putamen (Cosine distance) and Left-Hippocampus (Euclidean distance)—highlight the relevance of these regions in distinguishing PD stages. These regions have been previously implicated in PD-related neurodegeneration, reinforcing the validity of the feature selection process (Random Forest). For example, disruptions in the spatial relationships of the hippocampus have been linked to early cognitive decline in PD [76], while putamen volumetric reductions are closely associated with motor symptoms [77].

Similarly, changes in the caudate nucleus, including volumetric reductions and altered spatial relationships, have been linked to both motor and cognitive deficits in PD [78]. As a critical component of the striatal–thalamic circuitry disrupted by dopaminergic degeneration, the caudate plays a pivotal role in PD pathology [79]. This shows the importance and role of these deep gray matter structures as key biomarkers for PD stages.

Notably, parts of the glymphatic system, such as the 4th Ventricle and CSF spaces, were also identified as important features. The glymphatic system is essential for clearing waste from the brain, and dysfunction in this system has been increasingly associated with neurodegenerative diseases, including PD. Alterations in the spatial relationships and volumes of these structures may reflect impaired glymphatic flow, potentially contributing to the accumulation of pathological proteins, like tau and α-synuclein, in PD [80]. This highlights the importance of including glymphatic system components in analyses, as they may provide additional insights into the mechanisms driving PD progression.

An important strength of the proposed technique is that it relies on standard MRI sensors, rather than more specialized imaging techniques, like Diffusion Tensor Imaging (DTI), which is often used for explicit tractography mapping [81]. By using widely available standard MRI, the proposed approach has the potential for broader clinical applications, offering a more accessible and cost-effective method for detecting early neurodegenerative changes in PD. While DTI can provide detailed insights into white matter tracts, the ability to identify significant brain alterations using routine MRI presents the possibility of the integration of this method in regular clinical workflow.

### Limitations

It is important to acknowledge the limitations of this study. Although the models performed well, their generalizability to broader populations should be evaluated in more diverse cohorts. This study also only included cross-sectional data, and longitudinal follow-up could provide further insights into how these structural changes evolve over time.

Although a statistical analysis revealed no significant differences in age between the groups, and feature importance analysis indicated that age had a minimal impact on the models, and the potential influence of age-related changes in gray matter volume, particularly for participants in the prodromal group, cannot be entirely excluded. Healthy aging is known to accelerate gray matter loss after the age of 60, which could confound volumetric findings in specific tasks. Future studies should consider age as a covariate in regression models or apply further stratification by age to isolate disease-specific effects.

Furthermore, volumetric differences in the striatum and midbrain, as discussed by Ghaemi et al. (2002), highlight their potential in distinguishing Parkinson’s disease from multiple system atrophy. Including these regions in future studies using an extended brain atlas can further expand the diagnostic scope of this framework [82].

Despite these limitations, this research contributes to the body of evidence supporting the use of machine learning models for the early diagnosis and monitoring of PD [67,83]. The models presented the ability to effectively utilize volumetric and spatial feature data from standard MRI sensors that could potentially be integrated into clinical practice, providing non-invasive and interpretable diagnostic tools.

Moreover, to address these limitations and support further research, the Segmentation Workflow is available, which is an open-source project designed to conduct independent investigations using this technique. The Segmentation Workflow processes raw MRI data using FreeSurfer to extract parameters from specific regions of interest. These parameters, including volumes and centroids of selected brain structures, are aggregated, normalized, and exported to a CSV file, for data analysis and replication of the study’s methods. The workflow is publicly accessible on GitHub (please see Data Availability).

## 5. Conclusions

In conclusion, this study suggests that deep structure analysis can be used for the early detection of Parkinson’s disease and distinguishing between prodromal and fully diagnosed cases. These findings require further exploration into the clinical application of machine learning techniques for the early diagnosis of neurodegenerative diseases, which could eventually improve patient outcomes through earlier and more targeted interventions.

Moreover, this study challenges the current notion that automated volumetry is not informative, in the context of Parkinson’s disease, particularly in its prodromal stages [84]. The noted lack of informativeness may stem from the fact that traditional volumetric approaches do not incorporate spatial relationships, which are crucial for understanding the complex network topology of the brain.

By addressing the gap, this research proposes a novel method that integrates volumetric data with spatial metrics, specifically, Euclidean and Cosine distances relative to the thalamus. This combined approach could provide more sensitive and specific imaging markers, potentially improving the diagnostic and prognostic capabilities of MRI in Parkinson’s disease research, especially during the early stages of the disease.

## Figures and Tables

**Figure 1 sensors-24-08152-f001:**
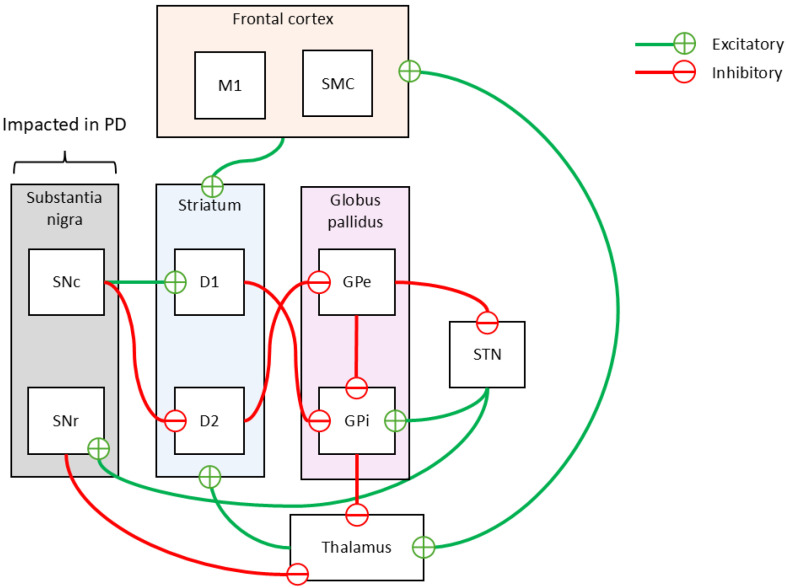
Schematic representation of basal ganglia circuitry and its disruption in Parkinson’s disease (PD). The diagram highlights the structures involved in movement regulation: the substantia nigra (SN), striatum (D1 and D2 pathways), globus pallidus externus (GPe) and internus (GPi), subthalamic nucleus (STN), thalamus, and frontal motor cortex (M1 and SMC). Green arrows indicate excitatory pathways, while red arrows represent inhibitory pathways. In PD, the loss of dopamine-producing neurons in the substantia nigra (SNc) disrupts the balance between the “GO” (direct) and “noGO” (indirect) pathways, leading to overactivation of the GPi and increased inhibition of the thalamus, eventually impairing movement signals to the motor cortex. Based on work by Przybyszewski et al. (2021) [20].

**Figure 2 sensors-24-08152-f002:**
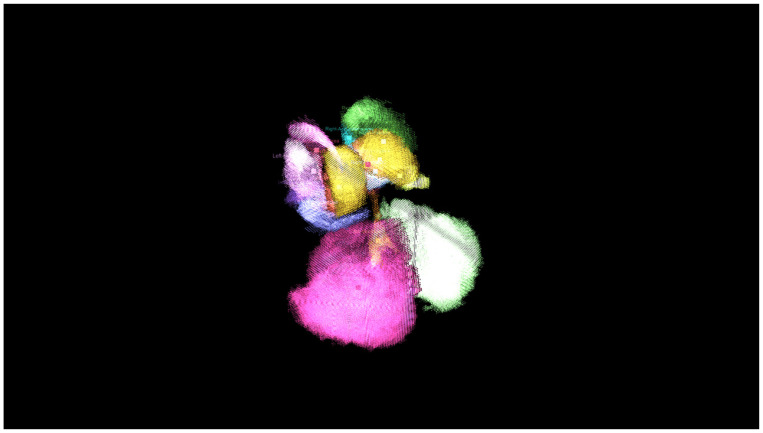
This 3D reconstruction illustrates the segmentation of selected brain regions of interest (ROIs) based on FreeSurfer processing. The segmented structures include the thalamus, hippocampus, amygdala, caudate, putamen, pallidum, cerebellum, and cerebrospinal fluid (CSF). Each structure is represented by a distinct color. In the center of each segmented structure, a central point (centroid) is displayed, representing the geometric center of that region.

**Figure 3 sensors-24-08152-f003:**
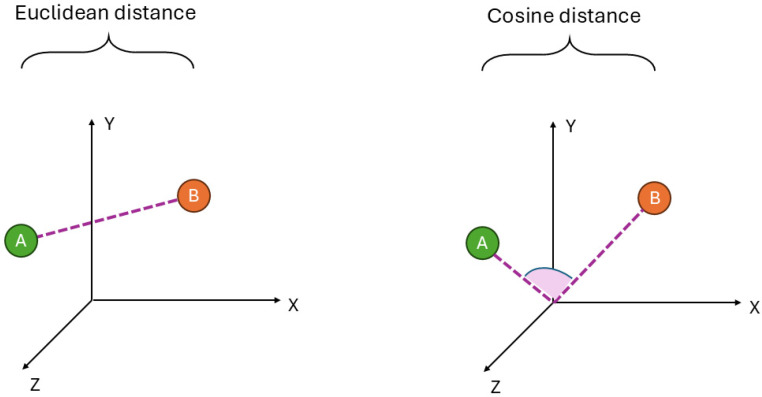
The Euclidean distance (considered as Minkowski distance with the parameter *p* = 2) represents the straight-line distance between the centroids of each structure (A) relative to the thalamus (B) in a three-dimensional space, providing a measure of physical separation. The Cosine distance is the angular difference between the vectors originating from the thalamus to the structures, showing their relative orientation in space.

**Figure 4 sensors-24-08152-f004:**
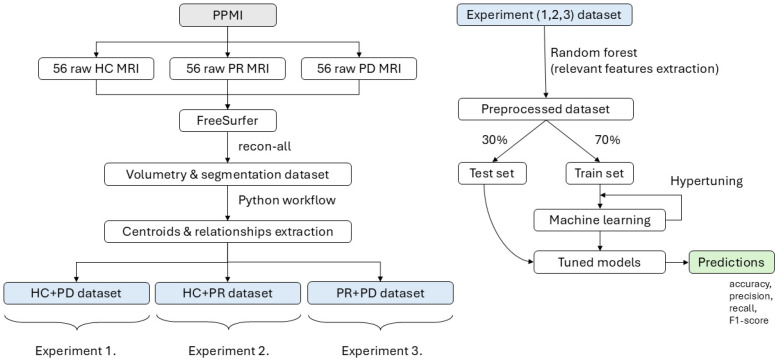
The workflow illustrates the processing pipeline starting from raw MRI data obtained from the PPMI dataset, including 56 healthy controls (HCs), 56 prodromal (PR), and 56 Parkinson’s disease (PD) subjects. The data are processed using FreeSurfer to extract volumetric and centroid-based features, which are then organized into three experimental datasets. In each experiment, Random Forest is applied for feature selection. This is followed by a 30/70 train–test split, hypertuning, and machine learning model training to generate predictions for each experiment.

**Figure 5 sensors-24-08152-f005:**
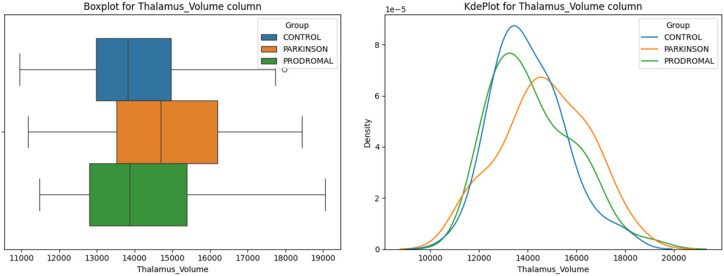
Thalamus volume distribution. The boxplot (**left**) shows the distribution of thalamus volumes for the control, Parkinson’s disease, and prodromal groups. The density plot (**right**) shows the kernel density estimates for the thalamus volumes in each group. The circles in the boxplot represent outliers beyond 1.5 times the IQR.

**Figure 6 sensors-24-08152-f006:**
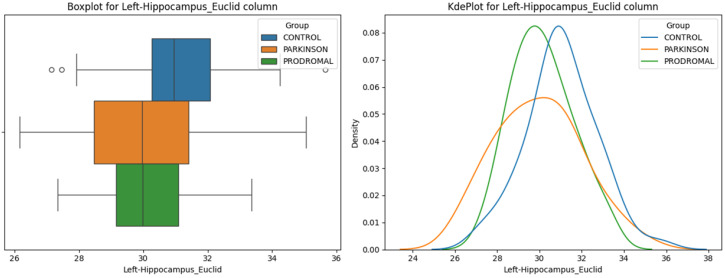
Left Hippocampus—Euclidean distance. The boxplot (**left**) shows the distribution of the normalized Euclidean distances between the left hippocampus and thalamus across the control, Parkinson’s disease, and prodromal groups. The density plot (**right**) illustrates the kernel density estimates of the distances for each group, showing variability in the distance between the left hippocampus and thalamus. The circles in the boxplot represent outliers beyond 1.5 times the IQR.

**Figure 7 sensors-24-08152-f007:**
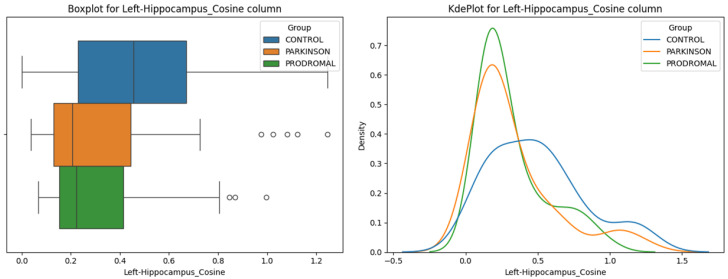
Left Hippocampus—Cosine distance. The boxplot (**left**) illustrates the distribution of Cosine distances between the left hippocampus and thalamus across the control, Parkinson’s disease, and prodromal groups. The density plot (**right**) shows the kernel density estimates of the Cosine distances for each group, showing variability in angular positioning of the left hippocampus with respect to the thalamus. The circles in the boxplot represent outliers beyond 1.5 times the IQR.

**Figure 8 sensors-24-08152-f008:**
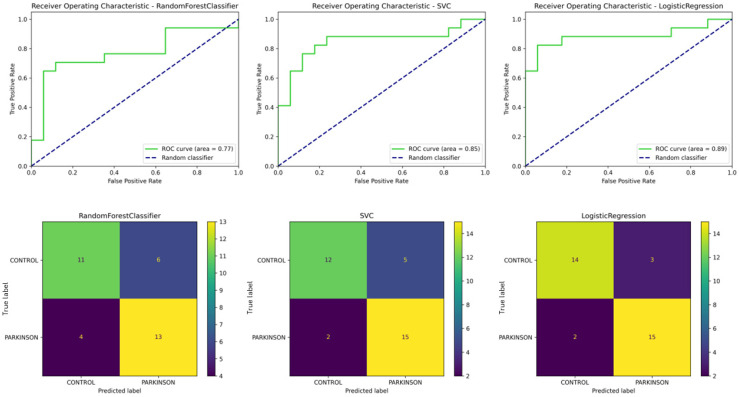
ROC curves and confusion matrices for classification of HC and PD. The top row shows the receiver operating characteristic (ROC) curves for the Random Forest (**left**), Support Vector Classifier (**middle**), and Logistic Regression (**right**) models. The area under the curve (AUC) values are presented for each model, with Logistic Regression achieving the highest AUC (0.89). The bottom row presents the corresponding confusion matrices for each model, illustrating the distribution of correctly and incorrectly classified participants for healthy controls and Parkinson’s patients. Logistic Regression achieved the best performance with 15 true positives and only 2 false negatives.

**Figure 9 sensors-24-08152-f009:**
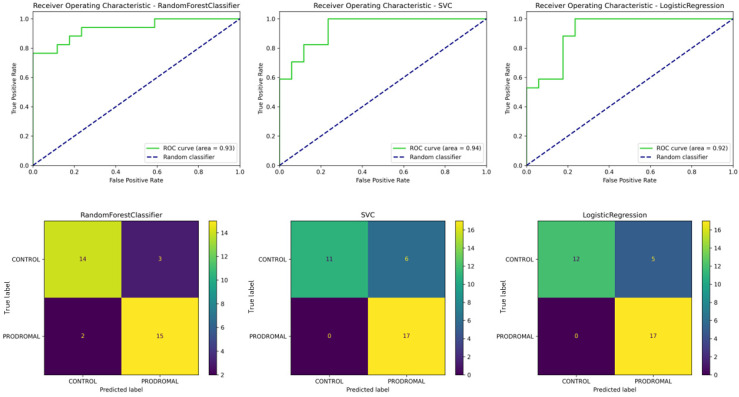
ROC curves and confusion matrices for classification of HC and PR. The top row shows the receiver operating characteristic (ROC) curves for the Random Forest (**left**), Support Vector Classifier (**middle**), and Logistic Regression (**right**) models. The area under the curve (AUC) values are presented for each model, with SVC achieving the highest AUC (0.99). The bottom row presents the corresponding confusion matrices for each model, illustrating the distribution of correctly and incorrectly classified participants for healthy controls and prodromal cases. The numbers in the figure represent the counts of correctly and incorrectly classified samples for each group. The Logistic Regression and SVC models showed the best performance, with both correctly classifying all prodromal cases and having only four and five false positives, respectively.

**Figure 10 sensors-24-08152-f010:**
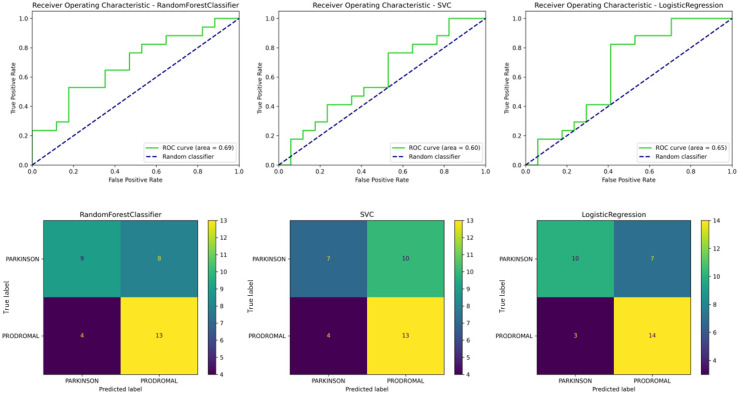
ROC curves and confusion matrices for classification of PD and PR. The top row shows the receiver operating characteristic (ROC) curves for the Random Forest (**left**), Support Vector Classifier (**middle**), and Logistic Regression (**right**) models. The area under the curve (AUC) values are presented for each model, with Logistic Regression achieving the highest AUC (0.73). The bottom row presents the corresponding confusion matrices for each model, illustrating the distribution of correctly and incorrectly classified participants for Parkinson’s disease and prodromal cases. The numbers in the figure represent the counts of correctly and incorrectly classified samples for each group. Logistic Regression showed the best performance, correctly classifying 15 prodromal cases and 9 Parkinson’s cases, while Random Forest and SVC demonstrated lower overall accuracy with more misclassifications.

**Table 1 sensors-24-08152-t001:** Parameters of selected MRI scans and sensors obtained from the PPMI study.

Parameter	Value
Sensor	MRI
Group	Control, Prodromal, Parkinson
Visit	Baseline (BL)
Acquisition Plane	Sagittal
Field Strength	3.0 Tesla
Slice Thickness	1.0 mm
Sequencing	3D T1-weighted or MPRAGE GRAPPA
Weighting	T1

**Table 2 sensors-24-08152-t002:** MRI Scanners and sequences used in the study.

Manufacturer	Model	Sequence	HC (n = 56)	PD (n = 56)	PR (n = 56)
Siemens	TrioTim	MPRAGE GRAPPA	46	0	0
Siemens	Verio	MPRAGE GRAPPA	9	0	0
GE	SIGNA Architect	SAG 3D T1-weighted	1	0	0
GE	DISCOVERY MR750	3D T1-weighted	0	5	10
GE	SIGNA Architect	3D T1-weighted	0	12	17
Philips	Achieva	3D T1-weighted	0	3	3
Philips	Achieva dStream	3D T1-weighted	0	34	26
Toshiba	Vantage Elan	3D T1-weighted	0	2	0

**Table 3 sensors-24-08152-t003:** The table presents the mean ± standard deviation for demographic and clinical parameters, including sex, age, and UPDRS3 scores, across the healthy control (CONTROL), Parkinson’s disease (PARKINSON), and prodromal (PRODROMAL) groups.

Parameter	*p*-Value	CONTROL (n = 56)	PARKINSON (n = 56)	PRODROMAL (n = 56)
Age, years	0.065	61.536 ± 11.492	61.839 ± 10.986	66.339 ± 6.162
Sex, female	0.260	0.321 ± 0.471	0.429 ± 0.499	0.250 ± 0.437
UPDRS	0.000	0.786 ± 1.670	23.893 ± 11.474	4.857 ± 5.910

**Table 4 sensors-24-08152-t004:** The table presents statistical comparisons (FDR-adjusted) between the volumes of selected brain structures across the healthy control (HC), Parkinson’s disease (PD), and prodromal (PR) groups. Bold values indicate statistically significant results.

Volume	ANOVA *p*-Value	HC vs. PD *p*-Value	HC vs. PR *p*-Value	PD vs. PR *p*-Value
Thalamus	0.108	0.062	0.793	0.167
Left-Hippocampus	0.065	0.035	0.363	0.227
Right-Hippocampus	0.589	0.338	0.710	0.674
**Left-Amygdala**	**0.001**	**0.002**	**0.002**	0.799
Right-Amygdala	0.945	0.839	0.721	0.945
**Left-Caudate**	**0.034**	**0.015**	0.096	0.465
Right-Caudate	0.200	0.089	0.184	0.873
**Left-Putamen**	**0.002**	**0.002**	**0.001**	0.984
**Right-Putamen**	**0.016**	**0.035**	**0.006**	0.707
Left-Pallidum	0.167	0.124	0.089	0.945
Right-Pallidum	0.945	0.799	0.859	0.945
**Left-Accumbens**	**0.001**	**0.001**	**0.001**	0.859
Right-Accumbens	0.347	0.257	0.184	0.963
**Left-VentralDC**	**0.003**	**0.002**	0.051	0.227
Right-VentralDC	0.133	0.065	0.168	0.710
4th-Ventricle	0.945	0.799	0.945	0.823
Left-Cerebellum	0.258	0.599	0.411	0.112
Right-Cerebellum	0.220	0.638	0.343	0.096
CSF	0.177	0.599	0.065	0.334

**Table 5 sensors-24-08152-t005:** The table presents statistical comparisons (FDR-adjusted) between the Euclidean distances of selected brain structures and the thalamus across the healthy control (HC), Parkinson’s disease (PD), and prodromal (PR) groups. Bold values indicate statistically significant results.

Euclidean Distance.	ANOVA *p*-Value	HC vs. PD *p*-Value	HC vs. PR *p*-Value	PD vs. PR *p*-Value
**Left-Hippocampus**	**0.015**	**0.015**	**0.019**	0.674
Right-Hippocampus	0.984	0.945	0.945	0.984
Left-Amygdala	0.184	0.411	0.051	0.477
Right-Amygdala	0.250	0.112	0.399	0.571
Left-Caudate	0.966	0.885	0.945	0.945
**Right-Caudate**	**0.015**	**0.011**	**0.017**	0.941
Left-Putamen	0.945	0.945	0.799	0.749
**Right-Putamen**	**0.017**	**0.006**	0.108	0.343
Left-Pallidum	0.721	0.475	0.629	0.889
**Right-Pallidum**	**0.000**	**0.000**	**0.004**	0.542
Left-Accumbens	0.945	0.945	0.803	0.920
Right-Accumbens	0.710	0.799	0.410	0.710
**Left-VentralDC**	**0.008**	0.072	**0.001**	0.389
Right-VentralDC	0.171	0.079	0.410	0.399
4th-Ventricle	0.436	0.312	0.945	0.363
**Left-Cerebellum**	**0.029**	**0.011**	0.244	0.206
Right-Cerebellum	0.107	0.062	0.707	0.178
CSF	0.051	0.095	0.018	0.674

**Table 6 sensors-24-08152-t006:** The table presents statistical comparisons (FDR-adjusted) between the **Cosine** distances of selected brain structures and the thalamus across the healthy control (HC), Parkinson’s disease (PD), and prodromal (PR) groups. Bold values indicate statistically significant results.

Cosine Distance	ANOVA *p*-Value	HC vs. PD *p*-Value	HC vs. PR *p*-Value	PD vs. PR *p*-Value
**Left-Hippocampus**	**0.009**	0.017	**0.011**	0.984
Right-Hippocampus	0.721	0.399	0.739	0.799
Left-Amygdala	0.171	0.362	0.071	0.478
Right-Amygdala	0.910	0.859	0.859	0.710
**Left-Caudate**	**0.011**	0.540	**0.001**	0.051
Right-Caudate	0.112	0.984	0.046	0.108
**Left-Putamen**	**0.004**	0.257	**0.000**	0.073
Right-Putamen	0.322	0.945	0.200	0.240
**Left-Pallidum**	**0.017**	0.411	**0.001**	0.108
Right-Pallidum	0.494	0.839	0.399	0.342
Left-Accumbens	0.128	0.945	0.051	0.117
Right-Accumbens	0.250	0.901	0.179	0.168
Left-VentralDC	0.220	0.362	0.066	0.674
Right-VentralDC	0.945	0.847	0.799	0.963
4th-Ventricle	0.187	0.095	0.701	0.244
Left-Cerebellum	0.343	0.312	0.945	0.167
Right-Cerebellum	0.242	0.762	0.311	0.094
CSF	0.342	0.410	0.296	0.479

**Table 7 sensors-24-08152-t007:** Experiment 1. The table presents a model comparison for HC vs. PD classification trained on 11 features (Euclidean distances of Right-Pallidum, Right-Putamen, Right-Caudate, Left-Hippocampus; Cosine distances of Left-VentralDC, Left-Putamen, Left-Hippocampus, Left-Pallidum; Volumes of Left-Amygdala, Left-Caudate, Left-Accumbens). The table includes the hyperparameters for each model and reports the accuracy, precision, recall, and F1 score for each. Logistic Regression achieved the highest performance with an accuracy of 0.85. The bold values in the table indicate the best-performing models.

Model Name	Accuracy	Precision	Recall	F1-Score
RandomForest	0.71	0.71	0.71	0.70
*• 75 decision trees, max. depth 3, min. 4 samples in each leaf, min. 2 samples to split.*				
SVC	0.79	0.80	0.79	0.79
*• Penalty factor of 10, a ’one-vs-one’ decision function, a gamma value of 0.01.*				
**LogisticRegression**	**0.85**	**0.85**	**0.85**	**0.85**
*• Penalty factor of 1.0 and runs up to 100 iterations to optimize its parameter.*				
RoughSets	0.82	0.81	0.81	0.81
*• 185 rules based on discernibility relationships between data attributes.*				

**Table 8 sensors-24-08152-t008:** Experiment 2. The table presents model a comparison for the HC vs. PR classification, trained on 16 features (Cosine distances of Left-Putamen, Left-Caudate, Left-Pallidum, Right-Pallidum, Left-VentralDC, Left-Hippocampus, Right-Caudate, Right-Putamen; Euclidean distances of Left-VentralDC, Left-Hippocampus, Right-Cerebellum, Right-Caudate; Volumes of Left-Accumbens, CSF, Left-Putamen, Left-Amygdala). The table includes the hyperparameters for each model and reports the accuracy, precision, recall, and F1 score for each. Logistic Regression achieved an accuracy of 0.85 and was slightly outperformed by Rough Sets (0.88 accuracy). The bold values in the table indicate the best-performing models.

Model Name	Accuracy	Precision	Recall	F1-Score
RandomForest	0.85	0.85	0.85	0.85
*• 75 decision trees, max. depth 3, min. 3 samples in each leaf, min. 2 samples to split.*				
SVC	0.82	0.87	0.82	0.82
*• Penalty factor of 10, a ’one-vs-one’ decision function, a gamma value of 0.01.*				
**LogisticRegression**	**0.85**	**0.89**	**0.85**	**0.85**
*• Penalty factor of 0.1 and runs up to 100 iterations to optimize its parameter.*				
**RoughSets**	**0.88**	**0.89**	**0.89**	**0.89**
*• 182 rules based on discernibility relationships between data attributes.*				

**Table 9 sensors-24-08152-t009:** Experiment 3. The table presents a comparison of models trained to differentiate between Parkinson’s disease (PD) and prodromal (PR) cases using 17 features (Cosine distances of Left-Putamen, Right-Accumbens, Left-Accumbens, Left-Cerebellum, Left-Amygdala, Left-Caudate, Right-Putamen, Right-Amygdala, 4th-Ventricle, Right-Caudate, Right-Cerebellum; Euclidean distance of Right-Cerebellum, Left-Hippocampus; Volumes of Left-Cerebellum, Left-Putamen, Left-VentralDC; and Age). The table includes the hyperparameters for each model and reports accuracy, precision, recall, and F1 score for each. Logistic Regression achieved the highest performance with an accuracy of 0.71. The bold values in the table indicate the best-performing models.

Model Name	Accuracy	Precision	Recall	F1-Score
RandomForest	0.65	0.66	0.65	0.64
*• 100 decision trees, max. depth 3, min. 3 samples in each leaf, min. 2 samples to split.*				
SVC	0.59	0.60	0.59	0.58
*• Penalty factor of 10, a ’one-vs-one’ decision function, a gamma value of 0.1.*				
**LogisticRegression**	**0.71**	**0.72**	**0.71**	**0.70**
*• Penalty factor of 0.1 and runs up to 100 iterations to optimize its parameter.*				
RoughSets	0.66	0.71	0.58	0.64
*• 271 rules based on discernibility relationships between data attributes.*				

## Data Availability

Data used in the preparation of this article were obtained on 15 July 2024 from the Parkinson’s Progression Markers Initiative (PPMI) database (www.ppmi-info.org/access-data-specimens/download-data (accessed on 15 July 2024)), RRID:SCR 006431. For up-to-date information on the study, visit www.ppmi-info.org (accessed on 15 July 2024). Because calculations of multiple MRIs are computationally extensive, the anonymous dataset used in this article is available in the “Appendix A”. Furthermore, to reproduce the steps required for this analysis, there are two open-source repositories available: “Segmentation Workflow” and “Classification Workflow”.

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
