# Peer review of "Machine Learning Recognizes Stages of Parkinson’s Disease Using Magnetic Resonance Imaging"

_sensors, 2024, doi:10.3390/s24248152_

Round 1

Reviewer 1 Report

Comments and Suggestions for Authors

Dear Authors,

First, I commend you on your efforts in this research.

Abstract:

e as digital biomarkers for early detection of PD, SUBSTITUTE: as neuroimaging

methods:

How many patients?

These scans were used for volumetric analysis and brain segmentation to investigate 119 structural differences across the groups, SUBSTUTUTE: volumetric

Discussion

This study demonstrates the utility of volumetric and spatial relationships of deep structures

SUBSTITUTE brain structures

You should include midbrain in the volumetric analysis and also include the following paper in this discussion:

Ghaemi M, WD. Differentiating multiple system atrophy from Parkinson's disease: contribution of striatal and midbrain MRI volumetry and multi-tracer PET imaging. J Neurol Neurosurg Psychiatry. 2002 Nov;73(5):517-23. 

INCLUDE IN DISCUSSION:

Since your goal was to find the best set of training data to improve classification, the information in excess can cause noise. This noise leads the model to learn random patterns that do not improve its ability to sort images correctly. This excess of information is called overfitting and is a well know problem in machine learning. Therefore, the best-ranked features can be used as a strategy to avoid overfitting and improve and optimize the model.

Alves AFF. Inflammatory lesions and brain tumors: is it possible to differentiate them based on texture features in magnetic resonance imaging? J Venom Anim Toxins Incl Trop Dis. 2020 Sep 4;26:e20200011. 

Author Response

Reviewer #1

---

Dear Reviewer,
I appreciate your detailed feedback on my manuscript. Your insights have been very valuable in refining this paper. Below, I address each of your points individually, explaining how I have considered them in revision process. All changes are marked in yellow in the manuscript, for better readability.

---

# Reviewer Comment 1: 
Abstract: as digital biomarkers for early detection of PD, SUBSTITUTE: as neuroimaging

# Author Response 1:
Thank you for this suggestion, which aligns with Reviewer #2's feedback on improving the clarity and focus of the abstract. I have revised the abstract to emphasize neuroimaging as the study’s focus, replacing "digital biomarkers" with precise descriptions of the neuroimaging features analyzed. The revised abstract now reads:

ABSTRACT: Neurodegenerative diseases (NDs) such as Alzheimer's disease (AD) and Parkinson's disease (PD) are debilitating conditions that affect millions worldwide, and the number of cases is expected to rise significantly in the coming years. Because early detection is crucial for effective intervention strategies, this study investigates whether structural analysis of selected brain regions, including volumes and their spatial relationships obtained from regular T1-weighted MRI scans (N=168, PPMI database), can model stages of PD using standard machine learning (ML) techniques. Thus, diverse ML models, including Logistic Regression, Random Forest, Support Vector Classifier, and Rough Sets, were trained and evaluated. Models used volumes, Euclidean, and Cosine distances of subcortical brain structures relative to the thalamus to differentiate between control (HC), prodromal (PR), and PD groups. Based on three separate experiments, the Logistic Regression approach was optimal, providing low feature complexity and high predictive accuracy (85%-88%) in PD stages recognition. Using interpretable metrics such as volume and centroid-based spatial distances, models achieved high diagnostic accuracy, presenting a promising framework for early-stage PD identification based on MRI scan.

Thank you for this opportunity to enhance the clarity of abstract.

# Reviewer Comment 2: 
Methods: How many patients?

# Author Response 2:
Thank you for pointing out the need for clarification regarding the sample size. I have updated the Methods section to include this information. The revised text now states:

MRI scans were acquired from control (n = 56), prodromal (n = 56), and Parkinson's disease (n = 56) groups during baseline (BL) visits, resulting in a total of 168 participants.

I appreciate your feedback, which has improved the clarity and completeness of the Methods section.

# Reviewer Comment 3:
Methods: These scans were used for volumetric analysis and brain segmentation to investigate 119 structural differences across the groups, SUBSTUTUTE: volumetric

# Author Response 3:
Thank you for this suggestion. I have revised the text in the Methods section accordingly. The updated sentence now reads:

"These scans were used for volumetric analysis and brain segmentation to investigate volumetric differences across the groups."

I appreciate your input in refining this section.

# Reviewer Comment 4:
Discussion: This study demonstrates the utility of volumetric and spatial relationships of deep structures: SUBSTITUTE brain structures

# Author Response 4:
Thank you for this valuable suggestion. I have updated the sentence in the Discussion section to improve clarity and alignment with the terminology. The revised text now states:

"This study demonstrates the utility of volumetric and spatial relationships of brain structures in distinguishing between healthy controls, prodromal, and Parkinson's disease patients. Importantly, this study highlights the value of both Euclidean and Cosine distances, together with normalized volumetric data, to assess brain structure changes in PD."

This revision enhances the overall clarity of the discussion.

# Reviewer Comment 5:
Discussion: You should include midbrain in the volumetric analysis 

# Author Response 5:
Thank you for this insightful suggestion. I have incorporated a discussion of the midbrain and referenced the recommended paper to enhance the depth and scope of the Discussion section. The revised text now includes:

"Furthermore, volumetric differences in the striatum and midbrain, as discussed by Ghaemi et al. (2002), highlight their potential in distinguishing Parkinson’s disease from multiple system atrophy. Including these regions in future studies using an extended brain atlas can further expand the diagnostic scope of this framework [82]."

I appreciate your recommendation, which has helped to enrich the discussion.

# Reviewer Comment 6:
INCLUDE IN DISCUSSION: Since your goal was to find the best set of training data to improve classification, the information in excess can cause noise. This noise leads the model to learn random patterns that do not improve its ability to sort images correctly. This excess of information is called overfitting and is a well-known problem in machine learning. Therefore, the best-ranked features can be used as a strategy to avoid overfitting and improve and optimize the model. Alves AFF. Inflammatory lesions and brain tumors: is it possible to differentiate them based on texture features in magnetic resonance imaging? J Venom Anim Toxins Incl Trop Dis. 2020 Sep 4;26:e20200011.

# Author Response 6:
Thank you for highlighting this important aspect of feature selection and overfitting. I have added the following statements to the Methods section under the "Machine Learning" subsection to address this point:

"The goal was to find the best set of training data to improve classification, but the information in excess can cause noise. This noise leads the model to learn random patterns that do not improve its ability to sort images correctly. This excess of information is called overfitting and is a well-known problem in machine learning. Therefore, the best-ranked features can be used as a strategy to avoid overfitting and improve and optimize the model [74]. This feature selection method follows findings from the study by Imran et al. (2018), where the authors applied five feature selection techniques (Gain Ratio, Kruskal-Wallis Test, Random Forest Variable Importance, RELIEF, and Symmetrical Uncertainty) along with four classification algorithms (K-Nearest Neighbor, Logistic Regression, Random Forest, and Support Vector Machine) on the PD dataset [75]. They found that Random Forest Variable Importance is one of the most impactful feature ranking techniques. Thus, in the present study, within each sub-dataset, Random Forest was employed first to vote for the most important features specific to each classification task."

I appreciate your suggestion to include the reference by Alves AFF, and I have incorporated it to strengthen the explanation of the overfitting problem and the role of feature selection in mitigating it.

---

Thank you again for your thorough and thoughtful feedback. I believe these amendments have strengthened the manuscript, providing a more nuanced understanding of the complexities in applying machine learning for the detection of neurodegenerative diseases. If there are any further areas requiring clarification, I would be happy to address them.

Reviewer 2 Report

Comments and Suggestions for Authors

This paper describes some potentially interesting methodology that could be relevant in the field of MRI-aided diagnosis. However, most of the methods are poorly motivated and implemented, some statistical methods are incorrect and the discussion is poor.  There are also many other issues which I discuss in detail below.

1.       Title: the title of this paper is inappropriate. The author performed binary comparisons between three groups (controls, prodromal and PD patients) and found that these could be accurately differentiated via the methods employed. Nothing in the work presented can claim to be predicting PD progression; this could only be claimed if longitudinal data was used. At the most, the author can claim his methods can differentiate between disease stages.

2.       Abstract: this could also be improved by removing all the unnecessary numerical detail, and instead using more space focusing on the merits/“take-home” messages of this study – currently a single sentence.

3.       Line 48: “But as the brain ages, long-distance connections tend to deteriorate faster than short-range connections.” – please provide a reference in support of this statement.

4.       Lines 71-77: The author points out some “gaps” or limitations in previous studies, however none of those are addressed by the present study. In particular, this sentence is rather puzzling: “Further more, studies on the connectome often incorporate additional subjective metrics, such as neuropsychological tests, to enhance the accuracy, sensitivity, and specificity of the findings, but this introduces variability that can make cross-study comparisons difficult.” I cannot be sure exactly which neuropsychological tests the author is referring to since no references are provided, but clinical questionnaires are often an integral part of the clinical diagnosis process, and comparing their performance against MRI based metrics is therefore essential to establish the usefulness of MRI metrics in this context. In general, the tests used in clinical practice have subjected to rigorous validation, with a vast number of studies in the literature demonstrating their value. The author’s concern that “this introduces variability that can make cross-study comparisons difficult” is somewhat baffling. Is the ultimate goal of this research to compare across studies, or to find the best features for ML-aided diagnosis/prognosis of PD?

5.       Lines 84-88: “To address the challenges of brain network analysis (…)”. The list of ROIs described in this paragraph seems completely arbitrary: thalamus, hippocampus, amygdala, caudate, putamen, pallidum, accumbens area, ventral diencephalon, cerebellum, fourth ventricle, and cerebrospinal fluid. Why have these ROIs been chosen?

6.       Line 94-95: “First, volumetric analysis will be performed on all structures. Following this, the centroids of selected structures will be used to calculate the Euclidean distance to the thalamus, providing a measure of their spatial separation” – again, no motivation is provided for why the thalamus has been chosen as a reference.

7.       Lines 102-104: “Unlike previous studies in the field, this approach offers a standardized and practical framework for PD research, which does not rely on subtle shape characteristics of brain structures or clinical tests.” – I completely disagree with this statement. The approach proposed by the author is just as arbitrary as any other! The choice of ROIs, using the thalamus as reference, the chosen distance metrics, etc, are not motivated anywhere and largely seem to reflect arbitrary choices/preferences by the author. And again, clinical tests are standardised and validated.

8.       Lines 115-120:  It seems that difference scanners and even different MRI sequences were used, however, the issues of data harmonization are completely ignored. Later on the author does state that freesurfer has been shown to be robust to different scanners/sequences (with no reference provided), but clearly not enough attention has been paid to this matter. Can the author please justify (using existing studies or otherwise), why it is valid to combine data com different scanners and sequences with no harmonisation methods applied. Can the author please also report the distribution of participants from each group across scanners. And uneven distribution could result in increased classification accuracy due to scanner differences, and therefore this issue needs to be investigated carefully.

9.       Lines 129-130: “Briefly, this processing includes motion correction and averaging of multiple volumetric T1 weighted images (when more than one is available)” – can the author please clarify exactly how many participants have repeated scans? Participants with repeated scans will have more accurate segmentations, which can again bias the classification results, especially if there is a higher number of multiple images per participant in one group (e.g. controls) vs the others.

10.   Lines 129: “Briefly, this processing includes motion correction” – can the author clarify what exactly was done to apply motion correction to the 3D T1-weighted data? This is extremely hard to do at the post-acquisition, post-reconstruction level and I am not aware of any methods to do so. Therefore more detail and references need to be provided.

11.   Lines 151-153: “Freesurfer morphometric procedures have been demonstrated to show good test-retest reliability across scanner manufacturers and across field strengths.” – related to point 9 above, please provide a reference and more specific relevant to justify why differences in scanner and sequence can just be ignored in the present study.

12.   Lines 156-157: recon-all is known to provide inaccurate reconstructions for some brains. The likelihood of incorrect reconstruction increases for older participants and patience populations. However, the author seemed to have performed no quality control or manual edits after recon-all. Again, this has the potential to introduce biases, given that recon-all is more likely to result in inaccuracies for patients when compared to controls. Please can the author clarify if any form of quality control was performed, and if not, please can this issue be given the attention it deserves. See for example https://surfer.nmr.mgh.harvard.edu/fswiki/FsTutorial/TroubleshootingData

13.   Lines 193-194: “In cases where volume measurements were not explicitly available from FreeSurfer, volumes were calculated based on voxel data.” – can the author please clarify exactly which ROIs had volumes estimated from freesurfer and which used another method. For the latter, either explain in more detail the method used to calculate the volume, or provide a reference.

14.   Lines 197-198: “All volumes of the deep structures and CSF were normalized per participant using their thalamus volume” – why was the volume of the thalamus used for normalisation, rather than the total intracranial volume which is routinely used for this purpose in the literature?

15.   Lines 205-207: “Group differences were evaluated using independent t-tests with unequal variance assumptions (Welch’s t-test), comparing HC vs PR, HC vs PD, and PR vs PD for each parameter.” – This approach is incorrect. An ANOVA test should be used to compare across the three groups, and only if that test is significant is it justified to perform the three pairwise post-hoc tests. This affects all results presented in tables 2 to 5. All tests need to be repeated and the findings re-interpreted.

16.   Lines 212-213: “P-values below 0.05 were marked as significant.” – in addition to the issue above, the author has carried out many t-tests but correction for multiple comparisons is not mentioned anywhere at all. This again affects all results in tables 2 to 5.

17.   Lines 207-209: “Additionally, the “Sex” variable was coded as binary, where male (M) was encoded as 0 and female (F) as 1, to allow for statistical comparison across groups.” – which test was used for this comparison? From the way the results are presented later it would seem that the author used also a t-test for this comparison, which is not appropriate. A chi-squared test should be used instead.

18.   Lines 225-227: “Within each sub-dataset, Random Forest was employed first to vote for the most important features specific to each classification task. Figure 3 shows the workflow with processing pipeline.” – please provide more detail about the feature selection used, including the thresholds used to select the most important features to be retained for further analysis. Can the author please also explain why this random forest approach was deemed to be the most appropriate method for feature selection for this particular project? At the moment this choice seem wholly arbitrary.

19.   Line 233: Given the small size of the dataset, can the author please justify why a 30/70 split was deemed appropriate? Why is cross-validation not performed?

20.   Lines 274-277: There is a significant difference in age between PR and the other groups. How can the author be sure this is not biasing any of the results? It is well known that healthy ageing alone will result in loss in grey-matter volume which accelerates after 60 years of age. Therefore how can we be sure that age alone cannot explain the differences observed between PR and controls? This needs to be addressed carefully and included in the discussion/limitations.

21.   Line 496: please provide more detail on the methods used for the Rough Set approach, and provide references.

22.   Discussion: In the discussion the author has made no effort to interpret the findings in the context of the disease under study. For example, there is no discussion on whether the ROIs selected by the feature selection method (random forest) have been previously found to be relevant in the context of PD, no discussion around the overlap between the features selected for the 3 experiments, etc.

23.   Lines 690-705: It is not clear whether the author did indeed perform multi-class classification using the ensemble technique described? If so can the methods description please be moved to the methods section and the results presented in the relevant section? The author refers to their code on Github so it seems this was indeed implemented, in which case I don’t understand why the outcome is not reported in the paper?

Author Response

Reviewer #2

---

Dear Reviewer,
I appreciate your detailed feedback on my manuscript. Your insights have been very valuable in refining this paper. Below, I address each of your points individually, explaining how I have considered them in revision process. All changes are marked in yellow in the manuscript, for better readability.

---

# Reviewer Comment 1:
Title: the title of this paper is inappropriate. The author performed binary comparisons between three groups (controls, prodromal and PD patients) and found that these could be accurately differentiated via the methods employed. Nothing in the work presented can claim to be predicting PD progression; this could only be claimed if longitudinal data was used. At the most, the author can claim his methods can differentiate between disease stages.

# Author Response 1:
Thank you for this insightful comment regarding the title. To better reflect the focus of this work, I have revised the title as follows.
Old Title: "Machine Learning Can Predict Parkinson's Disease Progression Using MRI"
New Title: "Machine Learning Recognizes Stages of Parkinson’s Disease Using MRI"

This new title accurately represents the findings, emphasizing that the methods differentiate between disease stages rather than predict disease progression. I appreciate your feedback.

# Reviewer Comment 2:
Abstract: this could also be improved by removing all the unnecessary numerical detail, and instead using more space focusing on the merits/“take-home” messages of this study – currently a single sentence.

# Author Response 2:
Thank you for this valuable feedback. I have revised the abstract to focus on the key take-home messages of the study, removing excessive numerical details while maintaining clarity about the study’s contributions. The updated abstract now reads:

ABSTRACT: "Neurodegenerative diseases (NDs) such as Alzheimer's disease (AD) and Parkinson's disease (PD) are debilitating conditions that affect millions worldwide, and the number of cases is expected to rise significantly in the coming years. Because early detection is crucial for effective intervention strategies, this study investigates whether structural analysis of selected brain regions, including volumes and their spatial relationships obtained from regular T1-weighted MRI scans (N=168, PPMI database), can model stages of PD using standard machine learning (ML) techniques. Thus, diverse ML models, including Logistic Regression, Random Forest, Support Vector Classifier, and Rough Sets, were trained and evaluated. Models used volumes, Euclidean, and Cosine distances of subcortical brain structures relative to the thalamus to differentiate between control (HC), prodromal (PR), and PD groups. Based on three separate experiments, the Logistic Regression approach was optimal, providing low feature complexity and high predictive accuracy (85%-88%) in PD stages recognition. Using interpretable metrics such as volume and centroid-based spatial distances, models achieved high diagnostic accuracy, presenting a promising framework for early-stage PD identification based on MRI scan."

# Reviewer Comment 3:
Line 48: “But as the brain ages, long-distance connections tend to deteriorate faster than short-range connections.” – please provide a reference in support of this statement.

# Author Response 3:
Thank you for pointing out the need to substantiate this statement. I have expanded the discussion and included relevant references to support the claim. The revised paragraph now reads:

"As the brain ages, long-distance connections tend to deteriorate faster than short-range connections [7, 8]. This 'disconnected brain' phenomenon, characterized by age-related reductions in structural and functional connectivity, has been shown to account for significant declines in executive function and processing speed. Specifically, structural connectivity changes, particularly in long-distance white matter tracts linking the prefrontal cortex to other regions, explain up to 82.5% of the decline in executive function over time [8]. Furthermore, this phenomenon is particularly evident in the structural connectome, where hubs—regions with long-distance, high-capacity connections and high metabolic rates—are more vulnerable to aging effects than peripheral connections [7]. These age-related changes in hub connections reflect biological vulnerability and also contribute to cognitive decline, such as reduced processing speed."

[7]    X. Li, A. Salami, and J. Persson, “Hub architecture of the human structural connectome: Links to aging and processing speed,” Neuroimage, vol. 278, p. 120270, Sep. 2023, doi: 10.1016/j.neuroimage.2023.120270.
[8]    A. M. Fjell, M. H. Sneve, H. Grydeland, A. B. Storsve, and K. B. Walhovd, “The Disconnected Brain and Executive Function Decline in Aging,” Cerebral Cortex, p. bhw082, Apr. 2016, doi: 10.1093/cercor/bhw082.

I appreciate your feedback, which allowed to improve the accuracy of this section.

# Reviewer Comment 4:
Lines 71-77: The author points out some “gaps” or limitations in previous studies, however none of those are addressed by the present study. In particular, this sentence is rather puzzling: “Furthermore, studies on the connectome often incorporate additional subjective metrics, such as neuropsychological tests, to enhance the accuracy, sensitivity, and specificity of the findings, but this introduces variability that can make cross-study comparisons difficult.” I cannot be sure exactly which neuropsychological tests the author is referring to since no references are provided, but clinical questionnaires are often an integral part of the clinical diagnosis process, and comparing their performance against MRI based metrics is therefore essential to establish the usefulness of MRI metrics in this context. In general, the tests used in clinical practice have subjected to rigorous validation, with a vast number of studies in the literature demonstrating their value. The author’s concern that “this introduces variability that can make cross-study comparisons difficult” is somewhat baffling. Is the ultimate goal of this research to compare across studies, or to find the best features for ML-aided diagnosis/prognosis of PD?

# Author Response 4:
Thank you for your detailed feedback. I recognize the need to elaborate on this point for greater clarity and have revised the Introduction section to address these concerns and provide appropriate references. The updated text now reads:

"These studies collectively demonstrate the potential of network analysis and geometric approaches in understanding neurodegenerative diseases and developing biomarkers for diagnosis and progression monitoring. 
However, multiple gaps remain that need to be addressed for practical implementation of these findings. First, there is no widely accepted optimal approach for defining nodes and edges that can be used for calculations, which leads to inconsistencies across studies in how brain regions and their connections are identified and modeled. Additionally, studies on the connectome often incorporate additional metrics, such as neuropsychological tests as co-variates, to enhance the accuracy, sensitivity, and specificity of the findings. 
Clinical questionnaires are often an integral part of the clinical diagnosis process, and comparing their performance against MRI based metrics is therefore essential to establish their usefulness. In general, the tests used in clinical practice have subjected to rigorous validation. Common choice includes MoCA that has proven to be more effective than some other cognitive tests, such as the Mini-Mental State Examination (MMSE), especially in the context of Parkinson’s disease [15], [16]. However, this introduces variability that can make selection of the best features for ML-aided prognosis difficult. This is because MoCA’s performance can vary across diverse cultural and educational backgrounds, potentially leading to misinterpretation of results [17]. Moreover, individuals with lower education levels may score lower on the MoCA, not necessarily due to cognitive impairment but due to the test’s design, which might favor those with higher educational and cultural backgrounds [18]. Another noteworthy concern is its susceptibility to practice effects, especially between the first and second administrations. This could potentially skew results and require consideration in clinical interpretation [19]. 
Furthermore, parameters determining the shape and structure of deep brain regions, which are critical for understanding neurodegenerative processes, are often influenced by the resolution limitations of MRI scanners, leading to potential inaccuracies in network characterization. These challenges created the need for the development of alternative approaches for defining nodes and edges in network models that can be applied consistently across different research settings."

[15]    C. Zadikoff et al., “A comparison of the mini mental state exam to the montreal cognitive assessment in identifying cognitive deficits in Parkinson’s disease,” Movement Disorders, vol. 23, no. 2, pp. 297–299, Jan. 2008, doi: 10.1002/mds.21837.
[16]    T. Smith, N. Gildeh, and C. Holmes, “The Montreal Cognitive Assessment: Validity and Utility in a Memory Clinic Setting,” The Canadian Journal of Psychiatry, vol. 52, no. 5, pp. 329–332, May 2007, doi: 10.1177/070674370705200508.
[17]    M. G. Borda, C. Reyes-Ortiz, M. U. Pérez-Zepeda, D. Patino-Hernandez, C. Gómez-Arteaga, and C. A. Cano-Gutiérrez, “Educational level and its Association with the domains of the Montreal Cognitive Assessment Test,” Aging Ment Health, vol. 23, no. 10, pp. 1300–1306, Oct. 2019, doi: 10.1080/13607863.2018.1488940.
[18]    G. Gagnon et al., “Correcting the MoCA for Education: Effect on Sensitivity,” Canadian Journal of Neurological Sciences / Journal Canadien des Sciences Neurologiques, vol. 40, no. 5, pp. 678–683, Sep. 2013, doi: 10.1017/S0317167100014918.
[19]    S. A. Cooley et al., “Longitudinal Change in Performance on the Montreal Cognitive Assessment in Older Adults,” Clin Neuropsychol, vol. 29, no. 6, pp. 824–835, Aug. 2015, doi: 10.1080/13854046.2015.1087596.

I appreciate your feedback, which allowed to clarify this section and strengthen the discussion by addressing the utility and limitations of neuropsychological tests in the context of ML-aided prognosis.

# Reviewer Comment 5:
Lines 84-88: “To address the challenges of brain network analysis (…)”. The list of ROIs described in this paragraph seems completely arbitrary: thalamus, hippocampus, amygdala, caudate, putamen, pallidum, accumbens area, ventral diencephalon, cerebellum, fourth ventricle, and cerebrospinal fluid. Why have these ROIs been chosen?

# Author Response 5:
Thank you for highlighting the need for clarity regarding the selection of regions of interest (ROIs). To address this, I have expanded the Introduction and Methods sections to explain the rationale behind the choice of these specific ROIs. Below is a summary of the revisions:

Introduction (New Section): "Disrupted Networks in Parkinson’s Disease" now states that PD is characterized by the degeneration of dopamine-producing neurons in the substantia nigra (SN), a key midbrain structure within the basal ganglia essential for movement control. The disruption of basal ganglia circuitry leads to imbalances in the “GO” (direct) and “noGO” (indirect) pathways, causing overactivation of the globus pallidus internus (GPi) and increased inhibition of the thalamus, impairing movement signals to the motor cortex. Additionally, the glymphatic system, responsible for the clearance of metabolic waste, may play a role in PD pathology. Its dysfunction, linked to impaired cerebrospinal fluid (CSF) dynamics, may contribute to neuroinflammation and substantia nigra degeneration.

These neurophysiological findings guided the selection of ROIs relevant to the pathophysiology of PD and the practical limitations of the segmentation tools used. The selected structures include those directly involved in motor regulation, such as the basal ganglia (caudate, putamen, pallidum), thalamus, and cerebellum, as well as those contributing to cognitive and emotional functions, such as the hippocampus and amygdala. Additionally, the inclusion of CSF allows for coarse evaluation of glymphatic system activity, which may influence neuroinflammatory processes in PD.

Methods (Feature Extraction Section): The ROIs were selected based on their relevance to PD pathophysiology and practical considerations, as follows:

Motor Control: The basal ganglia (caudate, putamen, pallidum), thalamus, and cerebellum play critical roles in motor regulation, compensating for the disrupted dopamine pathways in PD [50, 51]. Cognitive and Emotional Functions: The hippocampus and amygdala, part of the limbic system, are associated with cognitive decline and emotional disturbances in PD [52]. Reward and Motivation: The accumbens area, part of the ventral striatum, integrates motivational and reward signals, which can also be affected in PD [53]. Glymphatic System and Neuroinflammation: The inclusion of the fourth ventricle and CSF allows for assessment of glymphatic system function, which has been implicated in neurodegeneration [56].

[50]    N. R. McFarland and S. N. Haber, “Thalamic relay nuclei of the basal ganglia form both reciprocal and nonreciprocal cortical connections, linking multiple frontal cortical areas,” Journal of Neuroscience, vol. 22, no. 18, 2002, doi: 10.1523/jneurosci.22-18-08117.2002.
[51]    A. Charara, M. Sidibé, and Y. Smith, “Basal Ganglia Circuitry and Synaptic Connectivity,” in Surgical Treatment of Parkinson’s Disease and Other Movement Disorders, 2003. doi: 10.1385/1-59259-312-7:19.
[52]    T. J. van Mierlo, C. Chung, E. M. Foncke, H. W. Berendse, and O. A. van den Heuvel, “Depressive symptoms in Parkinson’s disease are related to decreased hippocampus and amygdala volume,” Movement Disorders, vol. 30, no. 2, 2015, doi: 10.1002/mds.26112.
[53]    I. N. Mavridis and E.-S. Pyrgelis, “Nucleus accumbens atrophy in Parkinson’s disease (Mavridis’ atrophy): 10 years later.,” Am J Neurodegener Dis, vol. 11, no. 2, 2022.
[56]    H. Benveniste, H. Lee, and N. D. Volkow, “The Glymphatic Pathway: Waste Removal from the CNS via Cerebrospinal Fluid Transport,” Neuroscientist, vol. 23, no. 5, pp. 454–465, 2017, doi: 10.1177/1073858417691030.

To ensure reproducibility, the study relied on FreeSurfer’s standard segmentation atlas, which provides a limited yet practical set of recognized structures. This approach balances anatomical specificity with computational feasibility. Future studies should consider incorporating more detailed atlases to capture additional regions, such as the substantia nigra, which could provide further insights into PD-related changes.

Furthermore, a schematic representation (Figure 1) illustrating the basal ganglia circuitry and its disruption in PD was added to the Introduction to visually support this discussion.

I believe these additions clarify the rationale behind the selection of ROIs and emphasize their relevance to PD pathology.

# Reviewer Comment 6:
Line 94-95: “First, volumetric analysis will be performed on all structures. Following this, the centroids of selected structures will be used to calculate the Euclidean distance to the thalamus, providing a measure of their spatial separation” – again, no motivation is provided for why the thalamus has been chosen as a reference.

# Author Response 6:
Thank you for pointing out the need to clarify the rationale for selecting the thalamus as the reference structure. I have expanded the discussion in the Methods section to explain this choice. The updated explanation now includes:

"All volumes of the deep brain structures and cerebrospinal fluid (CSF) were normalized per participant using their thalamus volume. Normalizing brain volumes to a reference is a standard technique to control for individual variability in brain size or shape, allowing comparisons between different groups (e.g., HC, PD, and PR) to focus on relative differences in the volumes of the structures of interest rather than being confounded by overall brain size differences. This normalization is particularly important in group studies, as it reduces inter-subject variability while minimizing artefacts introduced by normalization strategies."

The thalamus was chosen as the reference structure for the following reasons. The thalamus serves as a critical relay center for basal ganglia output to the cortex, making it highly relevant to Parkinson’s disease pathology and its associated motor and cognitive symptoms [50]. While intracranial volume (ICV) is a commonly used normalization metric, it includes cortical structures that were not the focus of this study. Normalizing to the thalamus allows the analysis to remain centered on subcortical structures directly implicated in PD. Among subcortical regions, the thalamus has consistently shown the highest positive correlation with ICV, regardless of the presence of neurodegenerative diseases such as Alzheimer’s disease (AD) [57]. This makes it a reliable surrogate for normalization in subcortical-focused studies. Thalamus-based normalization has recently been evaluated by Zhang et al. (2024), who found that it improves the detectability of hypoperfusion in Alzheimer’s disease while reducing artefacts associated with global mean normalization [58]. Following these findings, this study adopts thalamus normalization to enhance sensitivity and accuracy in distinguishing volumetric differences among PD stages.

[50]    N. R. McFarland and S. N. Haber, “Thalamic relay nuclei of the basal ganglia form both reciprocal and nonreciprocal cortical connections, linking multiple frontal cortical areas,” Journal of Neuroscience, vol. 22, no. 18, 2002, doi: 10.1523/jneurosci.22-18-08117.2002.
[57]    O. Voevodskaya et al., “The effects of intracranial volume adjustment approaches on multiple regional MRI volumes in healthy aging and Alzheimer’s disease,” Front Aging Neurosci, vol. 6, Oct. 2014, doi: 10.3389/fnagi.2014.00264.
[58]    L. X. Zhang, T. F. Kirk, M. S. Craig, and M. A. Chappell, “Thalamus normalisation improves detectability of hypoperfusion via arterial spin labelling in an Alzheimer’s disease cohort,” Aug. 14, 2024. doi: 10.1101/2024.08.13.24311671.

I believe this additional detail provides a clear and evidence-based justification for the use of the thalamus as the reference structure. Thank you for your suggestion, which has allowed me to clarify this critical methodological choice.

# Reviewer Comment 7:
Lines 102-104: “Unlike previous studies in the field, this approach offers a standardized and practical framework for PD research, which does not rely on subtle shape characteristics of brain structures or clinical tests.” – I completely disagree with this statement. The approach proposed by the author is just as arbitrary as any other! The choice of ROIs, using the thalamus as reference, the chosen distance metrics, etc, are not motivated anywhere and largely seem to reflect arbitrary choices/preferences by the author. And again, clinical tests are standardised and validated.

# Author Response 7:
Thank you for this critical feedback. I acknowledge the need to clarify and better motivate the choices made in the study to demonstrate that they are neither arbitrary nor lacking rationale. To address your concerns, I have removed the original statement and reframed the purpose of the study, providing a clear explanation of the methods and metrics employed. 

Below are the key changes made to the manuscript:

Introduction (New Section): Purpose of the Study
"This study evaluates whether data from standard MRI scans can be used to calculate relationships and volumes of selected brain structures can serve as straightforward parameters for structural relationship analysis. 
Therefore, this study focuses on interpretable metrics, namely the volumes and the spatial distances between the centroids of selected brain structures. These metrics should provide a simple representation of the brain’s topology, from which basic mathematical operations can be conducted to model parameters of connectivity and relationships. First, volumetric analysis will be performed on all structures. Following this, the centroids of selected structures will be used to calculate the Euclidean distance to the thalamus, providing a measure of their spatial separation. To capture more complex spatial relationships, a second metric, the Cosine distance, will be introduced, which accounts for the orientation of the structures relative to the thalamus. As a result, each structure will be characterized by three parameters: volume, Euclidean distance from the thalamus, and Cosine distance from the thalamus. The goal is to find the best features for ML-aided diagnosis of PD."

Methods (Revised Section): Feature Extraction – Spatial Metrics Computations
"The spatial metrics used in this study—Euclidean distance and Cosine distance—were selected based on their relevance to structural neuroimaging and multidimensional analysis. Euclidean Distance is a well-established metric in structural neuroimaging for measuring spatial relationships, the Euclidean distance aligns with the need to quantify physical displacements and structural changes in anatomical regions, particularly in the context of neurodegenerative diseases [59, 61]. Computation involves calculating the straight-line distance (in millimeters) between the centroids of brain structures and the thalamus. This was done in a 3D RAS (Right-Anterior-Superior) space using the FreeSurfer segmentation output."

"Cosine Distance is novel application in the context of brain structure analysis, Cosine distance evaluates angular relationships, providing complementary insights into the spatial orientation of brain structures relative to the thalamus. This metric has shown utility in various biomedical data analyses [60, 62, 63] and was experimentally investigated in this study for its potential to capture patterns of structural arrangement."

Regarding Thalamus as reference: "The thalamus was chosen due to its critical role in PD pathophysiology as a relay center for basal ganglia output to the cortex [50]. Thalamus-based normalization mitigates the potential confounding effects of overall brain size variability while maintaining focus on subcortical regions directly implicated in PD. Recent studies, such as Zhang et al. (2024), support the use of thalamus normalization to improve sensitivity in detecting structural changes [58]."

[50]    N. R. McFarland and S. N. Haber, “Thalamic relay nuclei of the basal ganglia form both reciprocal and nonreciprocal cortical connections, linking multiple frontal cortical areas,” Journal of Neuroscience, vol. 22, no. 18, 2002, doi: 10.1523/jneurosci.22-18-08117.2002.
[58]    L. X. Zhang, T. F. Kirk, M. S. Craig, and M. A. Chappell, “Thalamus normalisation improves detectability of hypoperfusion via arterial spin labelling in an Alzheimer’s disease cohort,” Aug. 14, 2024. doi: 10.1101/2024.08.13.24311671.
[59]    A. Garg, D. Lu, K. Popuri, and M. F. Beg, “Brain geometry persistent homology marker for Parkinson’s disease,” in 2017 IEEE 14th International Symposium on Biomedical Imaging (ISBI 2017), IEEE, Apr. 2017, pp. 525–528. doi: 10.1109/ISBI.2017.7950575.
[60]    M. Frasca and G. Tortora, “Visualizing correlations among Parkinson biomedical data through information retrieval and machine learning techniques,” Multimed Tools Appl, vol. 81, no. 11, pp. 14685–14703, May 2022, doi: 10.1007/s11042-021-10506-x.
[61]    J.-C. Delgado-González, J. Florensa-Vila, F. Mansilla-Legorburo, R. Insausti, and E. Artacho-Pérula, “Magnetic Resonance Imaging and Anatomical Correlation of Human Temporal Lobe Landmarks, in 3D Euclidean Space: A Study of Control and Alzheimer’s Disease Subjects,” Journal of Alzheimer’s Disease, vol. 57, no. 2, pp. 461–473, Mar. 2017, doi: 10.3233/JAD-160944.
[62]    P. Ghaderyan and S. M. Ghoreshi Beyrami, “Neurodegenerative diseases detection using distance metrics and sparse coding: A new perspective on gait symmetric features,” Comput Biol Med, vol. 120, p. 103736, May 2020, doi: 10.1016/j.compbiomed.2020.103736.
[63]    A. Ouhmida, A. Raihani, B. Cherradi, and O. Terrada, “A Novel Approach for Parkinson’s Disease Detection Based on Voice Classification and Features Selection Techniques,” International Journal of Online and Biomedical Engineering (iJOE), vol. 17, no. 10, pp. 111–130, Oct. 2021, doi: 10.3991/ijoe.v17i10.24499.

By removing the original statement and providing clear justifications for the study’s methodology, I believe the manuscript now addresses the concerns regarding arbitrariness. The updated text clarifies the rationale behind the choice of metrics, ROIs, and reference structures. Thank you for your feedback, which has strengthened the focus and transparency of this study.

# Reviewer Comment 8:
Lines 115-120:  It seems that difference scanners and even different MRI sequences were used, however, the issues of data harmonization are completely ignored. Later on the author does state that freesurfer has been shown to be robust to different scanners/sequences (with no reference provided), but clearly not enough attention has been paid to this matter. Can the author please justify (using existing studies or otherwise), why it is valid to combine data com different scanners and sequences with no harmonisation methods applied. Can the author please also report the distribution of participants from each group across scanners. And uneven distribution could result in increased classification accuracy due to scanner differences, and therefore this issue needs to be investigated carefully.

# Author Response 8:
Thank you for your detailed feedback. I have expanded the manuscript to address scanner variability, data harmonization, and participant distribution. The following additions have been made:

Methods: MRI Dataset Description, new text was added:
"PPMI is a multi-site study, and there was an uneven distribution of participants from each group across scanners. This variability could potentially affect classification accuracy due to differences in scanner characteristics such as software, sensors, algorithms, and scan protocols. Such heterogeneity poses challenges for machine learning (ML) analysis [27]."

"To mitigate these issues, FreeSurfer was chosen for data processing, as it has demonstrated good test-retest reliability across scanner manufacturers and field strengths [28, 29]. While this study did not employ additional harmonization methods, these can be incorporated in future studies to further improve the accuracy and generalizability of AI models [27]."

Results: MRI Scans and Sensors, new text was added:
A breakdown of scanners, sequences, and participant distribution was added (Table 2). Key acquisition parameters were standardized (e.g., 3D T1-weighted sequences, field strength 3.0 Tesla, voxel size 1 × 1 × 1 mm³). Steps were taken to mitigate variability, including using consistent protocols and FreeSurfer's automated processing pipeline, which is robust to inter-scanner differences.

To maintain balanced groups for ML analysis, 56 participants were included in each group (HC, PD, PR). Participants excluded due to motion artifacts or segmentation errors were removed to ensure data quality.

References supporting FreeSurfer’s robustness to scanner and sequence variability were included:

[28]    X. Han et al., “Reliability of MRI-derived measurements of human cerebral cortical thickness: The effects of field strength, scanner upgrade and manufacturer,” Neuroimage, vol. 32, no. 1, pp. 180–194, 2006.
[29]    M. Reuter, N. J. Schmansky, H. D. Rosas, and B. Fischl, “Within-Subject Template Estimation for Unbiased Longitudinal Image Analysis,” Neuroimage, vol. 61, no. 4, pp. 1402–1418, 2012, doi: 10.1016/j.neuroimage.2012.02.084.

These citations are from the official FreeSurfer methods citation guide (https://surfer.nmr.mgh.harvard.edu/fswiki/FreeSurferMethodsCitation).

I believe these revisions provide a clear justification for the methodological approach and address the reviewer’s concerns regarding scanner variability and data harmonization. Thank you for your valuable suggestions, which have strengthened this aspect of the study.

# Reviewer Comment 9:
Lines 129-130: “Briefly, this processing includes motion correction and averaging of multiple volumetric T1 weighted images (when more than one is available)” – can the author please clarify exactly how many participants have repeated scans? Participants with repeated scans will have more accurate segmentations, which can again bias the classification results, especially if there is a higher number of multiple images per participant in one group (e.g. controls) vs the others.

# Author Response 9:
Thank you for your detailed feedback. The following changes have been made:

"Process of segmentation includes motion correction and averaging [35] of multiple volumetric T1 weighted images (when more than one is available). However, in this study, no participants had repeated scans, and this method was not applied. The segmentation process also includes the removal of non-brain tissue using a hybrid watershed/surface deformation procedure [30], automated Talairach transformation, segmentation of the subcortical white matter and deep gray matter volumetric structures (including hippocampus, amygdala, caudate, putamen, ventricles) [32], [45] intensity normalization [37], tessellation of the gray matter white matter boundary, automated topology correction [36], [46], and surface deformation following intensity gradients to optimally place the gray/white and gray/cerebrospinal fluid borders at the location where the greatest shift in intensity defines the transition to the other tissue class [33], [38], [41]."

This clarification ensures that there was no bias introduced by repeated scans in any of the participant groups. Thank you for your suggestion, which has allowed me to provide a more detailed explanation of the segmentation process.

# Reviewer Comment 10:
Lines 129: “Briefly, this processing includes motion correction” – can the author clarify what exactly was done to apply motion correction to the 3D T1-weighted data? This is extremely hard to do at the post-acquisition, post-reconstruction level and I am not aware of any methods to do so. Therefore more detail and references need to be provided.

# Author Response 10:
Thank you for pointing this out. I have clarified this in the manuscript as follows. 

"Segmentation of brain regions includes motion correction and averaging [35] of multiple volumetric T1-weighted images (when more than one is available). However, in this study, no participants had repeated scans, and this method was not applied."

[35]    M. Reuter, H. D. Rosas, and B. Fischl, “Highly Accurate Inverse Consistent Registration: A Robust Approach,” Neuroimage, vol. 53, no. 4, pp. 1181–1196, 2010, doi: 10.1016/j.neuroimage.2010.07.020.

This ensures that no motion correction at the post-acquisition, post-reconstruction level was performed in this study. The reference provided aligns with the official FreeSurfer documentation. Thank you for allowing me to clarify this aspect further.

# Reviewer Comment 11:
Lines 151-153: “Freesurfer morphometric procedures have been demonstrated to show good test-retest reliability across scanner manufacturers and across field strengths.” – related to point 9 above, please provide a reference and more specific relevant to justify why differences in scanner and sequence can just be ignored in the present study.

# Author Response 11:
Thank you for highlighting this point. I have updated the manuscript to include the following references that demonstrate the robustness of FreeSurfer morphometric procedures across scanner manufacturers and field strengths:

"FreeSurfer morphometric procedures have been shown to exhibit good test-retest reliability across scanner manufacturers and field strengths [28, 29]."

[28]    X. Han et al., “Reliability of MRI-derived measurements of human cerebral cortical thickness: The effects of field strength, scanner upgrade and manufacturer,” Neuroimage, vol. 32, no. 1, pp. 180–194, 2006.
[29]    M. Reuter, N. J. Schmansky, H. D. Rosas, and B. Fischl, “Within-Subject Template Estimation for Unbiased Longitudinal Image Analysis,” Neuroimage, vol. 61, no. 4, pp. 1402–1418, 2012, doi: 10.1016/j.neuroimage.2012.02.084.

These references are aligned with the FreeSurfer documentation. Thank you for your suggestion.

# Reviewer Comment 12:
Lines 156-157: recon-all is known to provide inaccurate reconstructions for some brains. The likelihood of incorrect reconstruction increases for older participants and patience populations. However, the author seemed to have performed no quality control or manual edits after recon-all. Again, this has the potential to introduce biases, given that recon-all is more likely to result in inaccuracies for patients when compared to controls. Please can the author clarify if any form of quality control was performed, and if not, please can this issue be given the attention it deserves. See for example https://surfer.nmr.mgh.harvard.edu/fswiki/FsTutorial/TroubleshootingData

# Author Response 12:
Thank you for bringing up this important point. To address this concern, I have included a new section in the manuscript (Methods -> Data Preprocessing -> Quality Control) that outlines the quality control (QC) procedures implemented in this study. The added text is as follows:
"Quality Control: This study implemented a quality control pipeline to address the concerns regarding reconstructions accuracy. This workflow systematically evaluated the outputs of recon-all using automated and visual inspection tools. First, the pipeline scanned all study directories to identify subjects and their respective recon-all logs. It checked for critical errors and filtered relevant output to avoid false positives. For visual QC, the pipeline generated key anatomical screenshots (sagittal, coronal, axial) overlaid with segmentation (aseg.mgz) using Freeview, allowing general inspection of surface and segmentation accuracy. Additionally, aseg.stats files were parsed to summarize volumetric data, ensuring that structures are segmented correctly, and data was complete. Reconstruction relied strictly on automatic segmentation, without manual adjustment of recognized brain areas. Subjects with missing or empty files were flagged for review and excluded from the study (nine scans of Parkinson’s disease patients). Review document included the status of each subject, error logs, availability of statistical data, and scanner model metadata."

I believe this addition addresses the reviewer's concern about QC measures and highlights the steps taken to ensure data integrity. Thank you for pointing out this issue, as it allowed me to provide greater transparency regarding the preprocessing pipeline and data quality assessment.

# Reviewer Comment 13:
Lines 193-194: “In cases where volume measurements were not explicitly available from FreeSurfer, volumes were calculated based on voxel data.” – can the author please clarify exactly which ROIs had volumes estimated from freesurfer and which used another method. For the latter, either explain in more detail the method used to calculate the volume, or provide a reference.

# Author Response 13:
Thank you for your observation. I have clarified this point in the manuscript under the section:
"Methods -> Feature Extraction -> Volumetric Analysis: For all selected structures FreeSurfer provided explicit volume measurements, and these volumes were used directly in the analysis. All volumes of brain structures and CSF were normalized per participant using their thalamus volume."

This ensures that the volume data relied only on FreeSurfer's automated segmentation and measurement pipeline, without the need for alternative calculation methods. Thank you for highlighting this point.

# Reviewer Comment 14:
Lines 197-198: “All volumes of the deep structures and CSF were normalized per participant using their thalamus volume” – why was the volume of the thalamus used for normalisation, rather than the total intracranial volume which is routinely used for this purpose in the literature?

# Author Response 14:
Thank you for this insightful comment. I have added the following explanation to the manuscript under Methods -> Feature Extraction -> Volumetric Analysis:

"Volumetric Analysis: For all selected structures FreeSurfer provided explicit volume measurements, and these volumes were used directly in the analysis. All volumes of brain structures and CSF were normalized per participant using their thalamus volume. The idea of normalizing brain volumes to a reference is a technique to control individual variability in brain size or shape. This allows the comparisons between different groups (such as HC, PD, and PR) to focus on relative differences in the volumes of the structures of interest, rather than being confounded by overall brain size differences between individuals.
Data normalization is an important approach to reduce inter-subject variability in group studies, but care must be taken when choosing a normalization strategy to avoid introducing confounds or artefacts into the data In this study, normalization by the thalamus was chosen to focus on subcortical structures directly related to Parkinson’s disease pathology. While intracranial volume (ICV) is widely used, it includes cortical structures that are not relevant to this study.
However, it was found that among subcortical regions thalamus consistently presents the highest positive correlation with ICV, irrespective of the diagnosis of neurodegenerative disease (AD) [57]. Moreover, thalamus normalization method was recently investigated by Zhang et al. (2024) who observed that it improves the detectability of hypoperfusion in Alzheimer’s disease and lessens the artefacts caused by the commonly-used strategy of normalization using global mean [58]. Therefore, this study follows proposed strategies and applies normalization using thalamus volumetry."

[57]    O. Voevodskaya et al., “The effects of intracranial volume adjustment approaches on multiple regional MRI volumes in healthy aging and Alzheimer’s disease,” Front Aging Neurosci, vol. 6, Oct. 2014, doi: 10.3389/fnagi.2014.00264.
[58]    L. X. Zhang, T. F. Kirk, M. S. Craig, and M. A. Chappell, “Thalamus normalisation improves detectability of hypoperfusion via arterial spin labelling in an Alzheimer’s disease cohort,” Aug. 14, 2024. doi: 10.1101/2024.08.13.24311671.

Based on these considerations, normalization using thalamus volumetry was adopted to enhance the focus on relevant structures and minimize potential artefacts in the analysis. Thank you for the opportunity to clarify this point.

# Reviewer Comment 15:
Lines 205-207: “Group differences were evaluated using independent t-tests with unequal variance assumptions (Welch’s t-test), comparing HC vs PR, HC vs PD, and PR vs PD for each parameter.” – This approach is incorrect. An ANOVA test should be used to compare across the three groups, and only if that test is significant is it justified to perform the three pairwise post-hoc tests. This affects all results presented in tables 2 to 5. All tests need to be repeated and the findings re-interpreted.

# Author Response 15:
Thank you for pointing this out. I have reanalyzed all data using the recommended methodology and revised the manuscript accordingly. The updates include a detailed explanation in Methods -> Statistical Analysis and revised results throughout the manuscript.

Methods -> Statistical Analysis:
Descriptive statistics, including the mean and standard deviation, were calculated for each group across all relevant parameters. Group differences were evaluated using ANOVA as a global test to identify overall differences among groups. Post-hoc pairwise comparisons were subsequently conducted using Welch’s t-tests (independent t-tests with unequal variance assumptions) for the following group pairs: HC vs PR, HC vs PD, and PR vs PD.

To control for multiple comparisons and reduce the risk of Type I error, p-values were adjusted using the false discovery rate (FDR) correction. This approach was chosen for its balance between statistical rigor and sensitivity in high-dimensional datasets.

Additionally, the "Sex" variable was coded as binary (male [M] = 0, female [F] = 1) and analyzed across groups using chi-squared tests to account for its categorical nature.

Statistics were calculated for four categories of data: Demographics and clinical scores, Volumetric analysis, Cosine distances, Euclidean distances. All analyses were conducted using Python 3.10.12, with relevant libraries.

Updates to Results:

Demographics and Clinical Scores -> The "Sex" variable was analyzed using chi-squared tests, showing no significant differences in distribution across groups. ANOVA tests presented significant differences in UPDRS3 scores across groups.

Volumetric Analysis-> ANOVA tests identified significant differences in specific brain structures (e.g., amygdala, caudate, putamen) with post-hoc Welch’s t-tests confirming pairwise group differences where applicable.

Euclidean and Cosine Distances Analysis-> Euclidean and Cosine distances between the thalamus and selected brain regions were analyzed using ANOVA and post-hoc Welch’s t-tests, revealing significant spatial reorganizations in certain structures.

Full statistical outcomes, including ANOVA and post-hoc results, are detailed in Results. This updated methodology ensures compliance with the recommended statistical framework and provides a robust foundation for interpreting group differences. Thank you for your valuable suggestion, which has significantly improved the statistical rigor and reliability of the study’s findings.

# Reviewer Comment 16:
Lines 212-213-> “P-values below 0.05 were marked as significant.” – in addition to the issue above, the author has carried out many t-tests but correction for multiple comparisons is not mentioned anywhere at all. This again affects all results in tables 2 to 5.

# Author Response 16:
Thank you for pointing this out. To address this concern, I have updated the manuscript to include the following:

"To control for multiple comparisons and reduce the risk of Type I error, p-values were adjusted using the false discovery rate (FDR) correction. This method was selected for its balance between statistical rigor and sensitivity in high-dimensional datasets."

This adjustment has been applied to all relevant analyses, and the results have been updated accordingly in Tables. Thank you for bringing this to my attention.

# Reviewer Comment 17:
Lines 207-209: “Additionally, the “Sex” variable was coded as binary, where male (M) was encoded as 0 and female (F) as 1, to allow for statistical comparison across groups.” – which test was used for this comparison? From the way the results are presented later it would seem that the author used also a t-test for this comparison, which is not appropriate. A chi-squared test should be used instead.

# Author Response 17:
Thank you for highlighting this point. I have clarified in the manuscript that:

"Additionally, the "Sex" variable was coded as binary (male [M] = 0, female [F] = 1) and analyzed across groups using chi-squared tests to account for its categorical nature."

Thank you for bringing this to my attention, which has allowed me to refine the methodology.

# Reviewer Comment 18:
Lines 225-227: “Within each sub-dataset, Random Forest was employed first to vote for the most important features specific to each classification task. Figure 3 shows the workflow with processing pipeline.” – please provide more detail about the feature selection used, including the thresholds used to select the most important features to be retained for further analysis. Can the author please also explain why this random forest approach was deemed to be the most appropriate method for feature selection for this particular project? At the moment this choice seem wholly arbitrary.

# Author Response 18:
Thank you for this observation. I have expanded the explanation in the manuscript under Methods -> Machine Learning:

"The goal was to find the best set of training data to improve classification, but the information in excess can cause noise. This noise leads the model to learn random patterns that do not improve its ability to sort images correctly. This excess of information is called overfitting and is a well-known problem in machine learning. Therefore, the best-ranked features can be used as a strategy to avoid overfitting and improve and optimize the model [74]. 
This feature selection method follows findings of study by Imran et al. (2018) where authors applied five feature selection techniques namely: Gain Ratio, Kruskal-Wallis Test, Random Forest Variable Importance, RELIEF and Symmetrical Uncertainty along with four classification algorithms (K-Nearest Neighbor, Logistic Regression, Random forest, and Support Vector machine) on the PD dataset [75]. They found that Random Forest Variable Importance is one of the best impactful feature ranking techniques.
Thus, in present study within each sub-dataset, Random Forest was employed first to vote for the most important features specific to each classification task."

[74]    A. F. F. Alves et al., “Inflammatory lesions and brain tumors: is it possible to differentiate them based on texture features in magnetic resonance imaging?,” Journal of Venomous Animals and Toxins including Tropical Diseases, vol. 26, 2020, doi: 10.1590/1678-9199-jvatitd-2020-0011.
[75]    A. Al Imran, A. Rahman, H. Kabir, and S. Rahim, “The Impact of Feature Selection Techniques on the Performance of Predicting Parkinson’s Disease,” International Journal of Information Technology and Computer Science, vol. 10, no. 11, pp. 14–29, Nov. 2018, doi: 10.5815/ijitcs.2018.11.02.

Thank you for your suggestion, which has allowed me to provide better transparency.

# Reviewer Comment 19:
Line 233: Given the small size of the dataset, can the author please justify why a 30/70 split was deemed appropriate? Why is cross-validation not performed?

# Author Response 19:
Thank you for raising this point. I have clarified this in the manuscript:

"Existing research concluded that the optimum results are achieved if 20 to 30% of the data is used for model testing and the remaining 70 to 80% of the data is used to train the model [73]. Thus, a 30/70 train-test split was applied to ensure that 70% of the data was used for training and 30% for testing. This approach was chosen to maintain consistency across classification tasks and prioritize comparability of results."

Furthermore, cross-validation was incorporated during the training phase: "Additionally, hyperparameter tuning for LR, RF, and SVC was performed using grid search with cross-validation to optimize model performance. Figure 4 shows the workflow with processing pipeline."

[73]    A. Gholamy, V. Kreinovich, and O. Kosheleva, “Why 70/30 or 80/20 Relation Between Training and Testing Sets : A Pedagogical Explanation,” Departmental Technical Reports (CS), vol. 1209, 2018.

This approach balances the need for a consistent test set with the benefits of cross-validation during training. Thank you for raising this point.

# Reviewer Comment 20:
Lines 274-277: There is a significant difference in age between PR and the other groups. How can the author be sure this is not biasing any of the results? It is well known that healthy ageing alone will result in loss in grey-matter volume which accelerates after 60 years of age. Therefore how can we be sure that age alone cannot explain the differences observed between PR and controls? This needs to be addressed carefully and included in the discussion/limitations.

# Author Response 20:
Thank you for raising this important point. After recalculation using ANOVA, the results suggest that age is balanced across groups (p = 0.065). The updated demographic analysis is shown in Table 3.

Additionally, as noted in Experiment 3, feature importance analysis using the Random Forest model revealed that age had a low importance score (0.027439). It was only included in the classification model for the 'PARKINSON' vs 'PRODROMAL' task and was not utilized in other classification tasks, such as 'CONTROL' vs 'PARKINSON' or 'CONTROL' vs 'PRODROMAL.' This suggests that age did not play a significant role in driving the differences observed across groups.

To further address the potential confounding effect of age, I have included the following note in the manuscript:

Discussion -> Limitations:
"Although statistical analysis revealed no significant differences in age between the groups, and feature importance analysis indicated that age had minimal impact on the models, the potential influence of age-related changes in gray matter volume, particularly for participants in the prodromal group, cannot be entirely excluded. Healthy aging is known to accelerate gray matter loss after the age of 60, which could confound volumetric findings in specific tasks. Future studies should consider age as a covariate in regression models or apply further stratification by age to isolate disease-specific effects."

Thank you for raising this important point, which has allowed me to improve the discussion.

# Reviewer Comment 21:
Line 496: please provide more detail on the methods used for the Rough Set approach, and provide references.

# Author Response 21:
Thank you for raising this point. I have revised the manuscript to include the following explanation:

"For data modelling, four machine learning models were used, namely Logistic Regression (LR), Random Forest (RF), Support Vector Classifier (SVC), and Rough Sets (RS). The choice of LR, RF, and SVC algorithms is supported by their earlier usefulness in the context of neurodegenerative diseases detection [67]. In this study, LR, RF, and SVC were implemented in Python using the scikit-learn 1.5 (sklearn) library [68].
The Rough Set approach was included as an additional validation technique due to its demonstrated effectiveness in granular computing with rough-set rules [69]. This approach followed the Pawlak model for Rough Sets, which uses rule extraction and reasoning about data [70], [71], [72]. Parameters were optimized to minimize ambiguity and maximize coverage. Rough Sets were modeled in Rough Set Exploration System 2.2 (RSES) [70], [72]. "

[70]    Z. Pawlak, “Rough sets,” International Journal of Computer & Information Sciences, vol. 11, no. 5, pp. 341–356, Oct. 1982, doi: 10.1007/BF01001956.
[71]    Z. Pawlak, “Rough set theory and its applications,” Journal of Telecommunications and information technology, pp. 7–10, 2002.
[72]    Z. Pawlak, Rough sets: Theoretical aspects of reasoning about data, vol. 9. Springer Science & Business Media, 1991.

Thank you for highlighting this, which has allowed me to provide additional detail and references for the Rough Set methodology.

# Reviewer Comment 22:
Discussion: In the discussion the author has made no effort to interpret the findings in the context of the disease under study. For example, there is no discussion on whether the ROIs selected by the feature selection method (random forest) have been previously found to be relevant in the context of PD, no discussion around the overlap between the features selected for the 3 experiments, etc.

# Author Response 22:
Thank you for highlighting this. I have added the following to the discussion:
"Overlapping features across experiments—such as Left-Putamen (Cosine distance) and Left-Hippocampus (Euclidean distance)—highlight the relevance of these regions in distinguishing PD stages. These regions have been previously implicated in PD-related neurodegeneration, reinforcing the validity of the feature selection process (Random Forest). For example, disruptions in the spatial relationships of the hippocampus have been linked to early cognitive decline in PD [76], while putamen volumetric reductions are closely associated with motor symptoms [77]. 
Similarly, changes in the caudate nucleus, including volumetric reductions and altered spatial relationships, have been linked to both motor and cognitive deficits in PD [78]. As a critical component of the striatal-thalamic circuitry disrupted by dopaminergic degeneration, the caudate plays a pivotal role in PD pathology [79]. This shows the importance and role of these deep gray matter structures as key biomarkers for PD stages.
Notably, parts of the glymphatic system, such as the 4th Ventricle and CSF spaces, were also identified as important features. The glymphatic system is essential for clearing waste from the brain, and dysfunction in this system has been increasingly associated with neurodegenerative diseases, including PD. Alterations in the spatial relationships and volumes of these structures may reflect impaired glymphatic flow, potentially contributing to the accumulation of pathological proteins like tau and α-synuclein in PD [80]. This highlights the importance of including glymphatic system components in analyses, as they may provide additional insights into the mechanisms driving PD progression."

[76]    M. Gee, J. Dukart, B. Draganski, W. R. Wayne Martin, D. Emery, and R. Camicioli, “Regional volumetric change in Parkinson’s disease with cognitive decline,” J Neurol Sci, vol. 373, 2017, doi: 10.1016/j.jns.2016.12.030.
[77]    C. Owens-Walton et al., “Striatal changes in Parkinson disease: An investigation of morphology, functional connectivity and their relationship to clinical symptoms,” Psychiatry Res Neuroimaging, vol. 275, pp. 5–13, May 2018, doi: 10.1016/j.pscychresns.2018.03.004.
[78]    J. A. Grahn, J. A. Parkinson, and A. M. Owen, “The cognitive functions of the caudate nucleus,” Prog Neurobiol, vol. 86, no. 3, 2008, doi: 10.1016/j.pneurobio.2008.09.004.
[79]    J. Pasquini et al., “Clinical implications of early caudate dysfunction in Parkinson’s disease,” J Neurol Neurosurg Psychiatry, vol. 90, no. 10, 2019, doi: 10.1136/jnnp-2018-320157.
[80]    D. M. Lopes, S. K. Llewellyn, and I. F. Harrison, “Propagation of tau and α-synuclein in the brain: therapeutic potential of the glymphatic system,” Transl Neurodegener, vol. 11, no. 1, 2022, doi: 10.1186/s40035-022-00293-2.

Furthermore, the description in Introduction -> Disrupted Networks in Parkinson’s Disease, which details the GO/noGO pathways, further strengthens these connections by outlining the disrupted circuitry underlying PD pathology.

I hope this addresses the reviewer’s concerns by linking the selected features to established disease mechanisms and providing greater context for their relevance.

# Reviewer Comment 23:
Lines 690-705: It is not clear whether the author did indeed perform multi-class classification using the ensemble technique described? If so can the methods description please be moved to the methods section and the results presented in the relevant section? The author refers to their code on Github so it seems this was indeed implemented, in which case I don’t understand why the outcome is not reported in the paper?

# Author Response 23:
Thank you for raising this point. I have clarified the implementation and discussed it under Results -> Machine Learning -> Ensemble Technique Needs External Validation:
Ensemble Technique Needs External Validation
From a pragmatic perspective, it is feasible to merge the three best-performing models into a single workflow capable of receiving a single input and providing a unified diagnostic output. This approach employs an ensemble technique that aggregates the probabilistic outputs of individual models. By combining outputs from models trained on distinct classification tasks (HC vs PD, HC vs PR, PR vs PD), aggregated probabilities for each diagnostic category—control, prodromal, and Parkinson’s disease—can be derived.
In practice, the ensemble calculates the final prediction by selecting the class with the highest summed probability. Additionally, confidence levels can be categorized into tiers such as high, moderate, low, or extremely low, ensuring the output reflects both the predicted diagnosis and the certainty of the model. This methodology enhances robustness and interpretability, making it better suited for real-world clinical application.
However, this ensemble technique has not been validated on an independent dataset, which represents a limitation of this study. It is crucial to evaluate the ensemble on a completely separate test set that has not been used in training any of the models. Since the three models share overlapping training datasets (e.g., HC samples used in both HC vs PD and HC vs PR tasks), their probabilistic outputs may not be independent when validated on the same data. For future research, an independent test set with balanced samples of HC, PR, and PD categories should be used, ensuring no overlap with the training data. Such a dataset should also account for class imbalances common in real-world scenarios.

This addition explains the ensemble methodology and its limitations, highlighting the need for external validation in future work. Thank you for your observation, which has helped improve the clarity and comprehensiveness of this section.

---

Thank you again for your thorough and thoughtful feedback. Your detailed suggestions have significantly contributed to refining the methodology, enhancing the discussion, and improving the clarity and focus of this paper. I believe these amendments have strengthened the manuscript, providing a more nuanced understanding of the complexities in applying machine learning for the detection of neurodegenerative diseases. If there are any further areas requiring clarification, I would be happy to address them.